# Time-Consistent Robust Multi-Objective Reinforcement Learning via a Bellman–Isaacs Weight-Adversary Recursion

**Mingxi Hu** [1]   **Meiling Yu** [2]

## Abstract

Most multi-objective reinforcement learning (MORL) methods either condition on a fixed preference weight $w$ or consider episodic robustness where an adversary selects a single $w$ per episode. We study a time-consistent robustness model with reactive preferences: after each transition, an opponent chooses the next weight $w_{t+1}$ after observing $s_{t+1}$, and incurs a switching cost $\lambda D_\Phi(w_{t+1} \mid w_t)$ based on a Bregman divergence. This yields a Bellman–Isaacs recursion with an inner weight minimization at every backup. We prove the induced operator is a contraction and derive a Bellman-residual certificate that turns approximation error into a uniform bound on robust performance. We develop practical solvers in both tabular and deep settings using Bregman-prox inner updates and a stabilized fixed-point iteration. To evaluate robustness without optimistic critic reuse, we introduce BR-$K$, testing policies against $K$ independently trained best-response preference adversaries. Across MO-Gymnasium benchmarks, our approach consistently improves WRR under strong step-wise opponents over preference-conditioned baselines while keeping DRIFT smoothly controllable via $\lambda$.

## 1. Introduction

Real-world reinforcement learning problems rarely optimize a single metric. Robots trade off task success, energy, and safety; recommender systems balance engagement, diversity, and long-term satisfaction. Multi-objective reinforcement learning (MORL) captures this setting by assigning

each transition a vector reward $r_t \in \mathbb{R}^d$ and studying policies that manage trade-offs (Roijers et al., 2013; Hayes et al., 2022). A common operational choice is linear scalarization, $\langle w, r_t \rangle$, where a preference weight $w \in \mathcal{W}$ specifies how objectives are combined. Many modern MORL approaches treat $w$ as an exogenous input—learning policies or value functions conditioned on $w$ and aiming to cover a Pareto front or generalize across weights (Natarajan & Tadepalli, 2005; Abels et al., 2019; Yang et al., 2019; Reymond et al., 2022).

In deployment, however, the question is not only *which* $w$ to use but also *when* it is fixed. Preferences can be uncertain, state-dependent, or revised after outcomes materialize (e.g., switching toward safety after a near-miss) (Natarajan & Tadepalli, 2005; Abels et al., 2019). Examples include autonomous landing or mobile robotics, where a controller may normally balance progress, smoothness, and energy, but after a near-failure or low-battery event the effective priority can shift immediately toward safety or recovery. A natural robustness response is to optimize against preference uncertainty via a max–min criterion, typically with a single adversarial weight chosen at the beginning of the episode (Park et al., 2024; Byeon et al., 2025). But a one-shot choice cannot express state-reactive preference shifts, and it can be *time-inconsistent*: the weight that looks most adverse today need not remain most adverse after the system moves to a different region of the state space.

We study *time-consistent robust MORL* by modeling the preference weight as an *adversarial control* rather than a conditioning variable. After each transition, an opponent selects the next weight $w_{t+1}$ after observing $s_{t+1}$ (and, under our strong information structure, the realized reward), while paying a switching cost $\lambda D_\Phi(w_{t+1} \mid w_t)$ (Cesa-Bianchi et al., 2013) based on a Bregman divergence (Bregman, 1967). Intuitively, our model is a two-player game played along the trajectory. The agent chooses actions; an adversary chooses the preference weight that scores the vector reward. At time $t$ the agent picks $a_t$ in $s_t$, the environment yields $(s_{t+1}, r_t)$, and then the adversary selects $w_{t+1}$ after observing $s_{t+1}$ (and, under our information structure, $r_t$), paying a switching penalty for moving away from $w_t$. Decisions therefore occur in the augmented state $(s_t, w_t)$. This inter-

[1] School of Data Science, Fudan University, China [2] College of Artificial Intelligence, Nankai University, China. Correspondence to: Mingxi Hu <23110980005@m.fudan.edu.cn>, Meiling Yu <1120230250@mail.nankai.edu.cn>.

*Proceedings of the 43$^{rd}$ International Conference on Machine Learning*, Seoul, South Korea. PMLR 306, 2026. Copyright 2026 by the author(s).

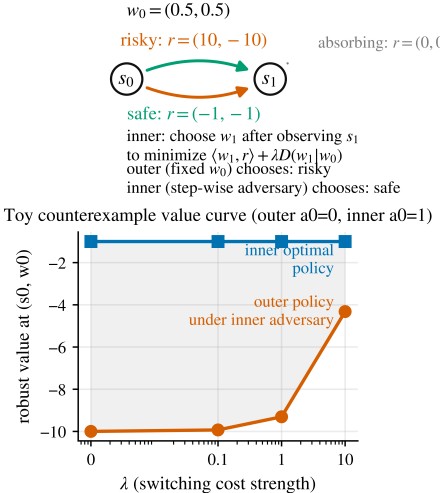

$w_0 = (0.5, 0.5)$

risky: $r = (10, -10)$       absorbing: $r = (0, 0)$

$s_0$       $s_1$

safe: $r = (-1, -1)$

inner: choose $w_1$ after observing $s_1$
to minimize $\langle w_1, r \rangle + \lambda D(w_1 | w_0)$
outer (fixed $w_0$) chooses: risky
inner (step-wise adversary) chooses: safe

Toy counterexample value curve (outer a0=0, inner a0=1)

*Figure 1.* A two-state counterexample showing that episode-level (outer) max–min over a fixed weight $w$ is not equivalent to the step-wise inner minimization induced by our formulation. The outer-optimal action is risky under $w_0 = (0.5, 0.5)$, but a step-wise opponent that selects $w_{t+1}$ after observing $s_{t+1}$ can switch weights to emphasize the weak objective, yielding a much lower robust value (orange) than the step-wise optimal safe policy (blue), across switching-cost strengths $\lambda$.

leaving explains the inner minimization inside each Bellman backup, which enforces time-consistent robustness. When $\lambda$ is large the adversary drifts; when small it switches sharply. This yields a discounted zero-sum stochastic game on the augmented state $(s_t, w_t)$ and induces a Bellman–Isaacs recursion with an *inner* minimization over $w_{t+1}$ inside each Bellman backup (Shapley, 1953; Littman, 1994; Iyengar, 2005; Ruszczyński, 2010). Placing the minimization inside the backup enforces time consistency: the continuation game from $(s_{t+1}, w_{t+1})$ has the same structure as from $(s_t, w_t)$, so the robust value is governed by a single fixed point rather than an outer minimizer that can become dynamically inconsistent.

Figure 1 isolates the difference between episodic (outer) and step-wise (inner) robustness in the smallest setting we could fit on one column: two states, two actions, and two reward coordinates. With initial weight $w_0 = (0.5, 0.5)$, the episodic objective selects the risky action, but a step-wise opponent can react after the transition by choosing $w_{t+1}$ as a function of $s_{t+1}$ (and, under the strong information structure, the realized reward). By switching emphasis to the most adverse objective, the opponent drives the risky branch to a much lower robust value. Consequently, placing the minimization inside the Bellman backup can change which action is robustly optimal even in a two-step MDP.

Turning this formulation into a practical learning and evaluation pipeline raises two bottlenecks. First, the inner minimization couples the immediate scalarization and the continuation value, so the minimizer depends on the current critic

and changes during learning; even when $\mathcal{W}$ is a simplex, the minimizer is generally nontrivial under function approximation (Bertsekas & Tsitsiklis, 1996). Second, robust evaluation is easy to understate because training a strong step-wise opponent is itself a nonconvex RL problem (Pinto et al., 2017): single-run opponents can be weak, and reusing the agent's critic to score the opponent can introduce optimism.

These challenges make the problem nontrivial even when the underlying MDP is simple: the robust objective is defined by a moving inner $\arg\min$ that depends on the critic, and naïvely caching adversary actions breaks the fixed point. Our solution keeps the game structure explicit. In the tabular regime we discretize $\mathcal{W}$ and run value iteration on $(s, w)$, producing a Bellman-residual certificate that lets us audit approximate solvers. In the deep regime we learn $Q_\theta(s, w, a)$ while recomputing the inner minimizer on-the-fly using a few mirror/prox steps (Beck & Teboulle, 2003) (or enumeration on a small grid), so the target remains aligned with the Bellman–Isaacs operator. Finally, to make robustness claims comparable across methods, we evaluate every learned policy against a separately trained step-wise best-response adversary and report a conservative multi-start aggregation (BR-$K$).

**Contributions.** We make four contributions (highlighting what is new in this paper): (i) **Objective:** we formalize time-consistent robustness in MORL by modeling $w$ as a step-wise adversarial control with a Bregman switching cost, and relate the resulting recursion to a dynamic preference–risk operator (Appendix A); (ii) **Theory:** we characterize the induced Bellman–Isaacs operator—proving $\gamma$-contraction, establishing stationary equilibria (for finite grids and, under mild regularity, for continuous compact $\mathcal{W}$), and deriving auditable certificates that convert Bellman residuals into robust-performance guarantees; (iii) **Algorithms:** we develop practical solvers that target the same fixed point in both grid/tabular and deep regimes, using online inner minimization embedded in value iteration and deep fixed-point learning; and (iv) **Evaluation:** we introduce BR-$K$, a method-agnostic multi-restart best-response evaluation protocol, and empirically demonstrate improved WRR under strong step-wise opponents with a controllable WRR–DRIFT tradeoff via $\lambda$.

## 2. Related Work

Multi-objective reinforcement learning (MORL) studies sequential decision making with vector-valued rewards and has traditionally emphasized Pareto-optimality, coverage of trade-off fronts, and evaluation of sets of solutions rather than a single scalar objective (Roijers et al., 2013; Hayes et al., 2022; Vamplew et al., 2011). Much of the deep MORL literature learns policies or value functions conditioned on an explicit preference weight and targets general-

ization across weights, including dynamic-weight schemes and coverage-oriented architectures (Natarajan & Tadepalli, 2005; Abels et al., 2019; Yang et al., 2019; Reymond et al., 2022). Our setting is complementary: the weight is not an exogenous input but an adversarial control that can react to realized transitions.

Robustness in MORL is often formalized through an episodic max–min criterion, where an adversary selects a single weight for the episode and the agent maximizes the corresponding robust weighted return (Park et al., 2024; Byeon et al., 2025). By contrast, we place the minimization inside the Bellman backup, yielding a time-consistent recursion that permits state-reactive preference switches. This differs from classical robust MDP and distributionally robust RL, which typically adversarialize transition kernels or data distributions rather than preferences (Iyengar, 2005; Nilim & El Ghaoui, 2005; Lu et al., 2024).

Our formulation can be viewed as a discounted zero-sum stochastic game on the augmented state $(s, w)$, governed by a Bellman–Isaacs (Shapley) operator (Shapley, 1953; Littman, 1994; Feng et al., 2024). The use of a nested (recursive) operator as the objective-defining mechanism is also reminiscent of time-consistent risk-sensitive control and dynamic risk measures (Ruszczyński, 2010; Tamar et al., 2015).

Beyond preference robustness, our work is also connected to the offline evaluation and safe-deployment literature, where one seeks reliable performance guarantees from fixed data. High-confidence and doubly-robust off-policy evaluation tools enable conservative deployment decisions and safe policy improvement (Thomas et al., 2015; Jiang & Li, 2016), including robust-baseline regret criteria (Petrik et al., 2016) and bootstrapping-based constraints (Laroche et al., 2019). Complementary pessimistic / conservative offline RL methods aim to lower-bound value under distribution shift by penalizing out-of-data actions or uncertain states (Fujimoto et al., 2019; Kumar et al., 2020; Kidambi et al., 2020; Jin et al., 2021), and benchmark suites such as D4RL (Fu et al., 2020) have accelerated empirical comparisons. Finally, adversarial training in RL has been explored via explicit two-player games over disturbances (Pinto et al., 2017), which is complementary to our preference-adversary game.

# 3. Problem Formulation

With linear scalarization, fixing a preference weight $w$ reduces MORL to an ordinary MDP that optimizes the discounted weighted return. Preference uncertainty is often handled by taking a max–min over a single weight chosen once per episode, but such an outer minimization cannot react to the realized next state and can become dynamically inconsistent when the most adverse direction changes af-

ter a transition. We instead treat the weight sequence as an adversarial control updated after each transition, while charging an explicit switching cost to keep preference drift interpretable and bounded. This yields a discounted zero-sum game on the augmented state $(s, w)$ whose equilibrium value is characterized by a Bellman–Isaacs recursion.

## 3.1. Multi-objective MDP and preference weights

Let $\mathcal{S}$ be a state space and $\mathcal{A}$ an action space. The environment dynamics are given by a transition kernel $P(\cdot \mid s, a)$ and a bounded vector reward function $r : \mathcal{S} \times \mathcal{A} \times \mathcal{S} \to \mathbb{R}^d$. For a trajectory $(s_t, a_t)_{t \geq 0}$, define the discounted vector return $R \triangleq \sum_{t \geq 0} \gamma^t r(s_t, a_t, s_{t+1})$ with discount factor $\gamma \in (0, 1)$. Note that rewards depend on the realized transition $(s_t, a_t, s_{t+1})$, which will matter for the opponent observation model defined below. Preferences are represented by weights $w \in \mathcal{W} \subseteq \mathbb{R}^d$, where $\mathcal{W}$ is a compact convex subset of the probability simplex $\{w \in \mathbb{R}^d_+ : \sum_i w_i = 1\}$. Let $\Phi : \mathcal{W} \to \mathbb{R}$ be a differentiable, 1-strongly convex function (a mirror map). The associated Bregman divergence is

$$D_\Phi(w' \mid w) \triangleq \Phi(w') - \Phi(w) - \langle \nabla\Phi(w), w' - w \rangle.$$

We instantiate this switching cost with KL divergence (negative entropy mirror map) and squared $\ell_2$ divergence (Euclidean mirror map); closed-form inner updates are given in Appendix C.5. A scalar $\lambda \geq 0$ controls the strength of the switching cost.

## 3.2. A step-wise preference adversary

The game state is the pair $(s_t, w_t)$. Given $(s_t, w_t)$, the agent samples an action $a_t \sim \pi(\cdot \mid s_t, w_t)$. The environment transitions to $s_{t+1} \sim P(\cdot \mid s_t, a_t)$ and reveals the vector reward $r_t = r(s_t, a_t, s_{t+1})$. The opponent then observes an information tuple $z_t$ (defined below) before choosing the next weight $w_{t+1} \sim \sigma(\cdot \mid z_t)$. The opponent acts after observing the transition outcome, hence selects $w_{t+1}$ to scalarize the just-realized reward $r_t$.

This order of moves is crucial: the opponent can condition its update on the realized next state, so the minimizing choice generally cannot be fixed in advance at the start of an episode. The next game state becomes $(s_{t+1}, w_{t+1})$, which preserves the Markov structure and leads to a one-step recursion with an inner minimization over $w_{t+1}$. The scalar stage payoff (agent reward) is

$$g_t \triangleq \langle w_{t+1}, r_t \rangle + \lambda\, D_\Phi(w_{t+1} \mid w_t).$$

The agent seeks to maximize the expected discounted sum of $(g_t)$, while the opponent seeks to minimize it. We consider two variants that differ only in what the opponent observes before choosing $w_{t+1}$.

**Weak opponent (state–weight):** the opponent observes $z_t^{\text{weak}} = (s_{t+1}, w_t)$.

Strong opponent (state–weight–reward): the opponent observes $z_t^{\text{strong}} = (s_{t+1}, w_t, r_t)$, equivalently $(s_t, a_t, s_{t+1}, w_t)$ when $r_t = r(s_t, a_t, s_{t+1})$ is deterministic given the transition.

Throughout, the Bellman–Isaacs operator and our robustness claims use the strong opponent; the weak opponent is only used for sanity-check baselines. Because the opponent acts after the transition is realized, the scalarization weight can react to information revealed at that time. Under the strong opponent (or whenever $r_t$ is deterministic given the transition), the opponent effectively has access to $r_t$ before selecting $w_{t+1}$, so using $\langle w_{t+1}, r_t \rangle$ models an "ex post" preference shift while keeping the continuation game Markov in $(s_{t+1}, w_{t+1})$.

### 3.3. Value and Bellman–Isaacs recursion

For an initial pair $(s, w)$, define the value under agent policy $\pi$ and opponent policy $\sigma$ as

$$J^{\pi,\sigma}(s,w) \triangleq \mathbb{E}_{s_0=s,\, w_0=w} \Big[ \sum_{t \geq 0} \gamma^t \big( \langle w_{t+1}, r_t \rangle \\ + \lambda D_\Phi(w_{t+1} \mid w_t) \big) \Big].$$

The robust value is the max–min equilibrium value

$$V^\star(s,w) \triangleq \max_\pi \min_\sigma J^{\pi,\sigma}(s,w).$$

This is a discounted zero-sum stochastic game on $(s, w)$, and $V^\star$ is its equilibrium value.

Unless stated otherwise, this recursion corresponds to the strong opponent (state-weight-reward), equivalently any setting where the opponent can infer $r(s, a, s')$ from its observation. Let $\mathcal{V}$ be the space of bounded real-valued functions on $\mathcal{S} \times \mathcal{W}$. Define the Bellman–Isaacs operator $\mathcal{T} : \mathcal{V} \to \mathcal{V}$ by

$$(\mathcal{T}V)(s,w) \triangleq \max_{a \in \mathcal{A}} \mathbb{E}_{s' \sim P(\cdot|s,a)} \Big[ \min_{w' \in \mathcal{W}} \big\{ \langle w', r(s,a,s') \rangle \\ + \lambda D_\Phi(w' \mid w) + \gamma V(s', w') \big\} \Big] \quad (1)$$

The defining feature is the inner minimization over $w'$ inside the one-step backup; it is this nesting that makes the objective time-consistent. Everything that follows—properties, algorithms, and evaluation—targets the fixed point of this operator.

*Time-consistency view.* For any bounded continuation slice $X : \mathcal{W} \to \mathbb{R}$ and realized tuple $(s', w, r)$, define

$$\rho_{s',w,r}(X) \triangleq \min_{w' \in \mathcal{W}} \{ \langle w', r \rangle + \lambda D_\Phi(w' \mid w) + \gamma X(w') \}.$$

Then $(\mathcal{T}V)(s,w) = \max_a \mathbb{E}_{s'}[\rho_{s',w,r(s,a,s')}(V(s',\cdot))]$. The mapping $\rho$ is monotone, satisfies discounted translation

equivariance, and is $\gamma$-Lipschitz in $X$, placing our objective in the standard nested one-step form for time-consistent dynamic risk evaluation (Ruszczyński, 2010); see Appendix A.

An episode-level max–min baseline chooses a single weight $w$ at the beginning of an episode:

$$V^{\text{outer}}(s) \triangleq \max_\pi \min_{w \in \mathcal{W}} \mathbb{E}_{s_0=s} \Big[ \sum_{t \geq 0} \gamma^t \langle w, r_t \rangle \Big].$$

Even when a switching penalty is introduced heuristically, an episode-level minimization cannot express state-dependent preference shifts and does not induce the recursion in (1). Figure 1 demonstrates that the two objectives can disagree on which action is robustly optimal. More fundamentally, the outer minimizer need not be dynamically consistent with the continuation game after the state changes, hence this criterion is not time-consistent.

We also consider an episodic adversary that chooses a single weight per episode (with an episode-level switching penalty) as a baseline for non-reactive preference uncertainty.

## 4. Bellman–Isaacs Recursion and Core Properties

This section records the core properties that make (1) a usable learning target. We emphasize them because they justify (i) value iteration and deep fixed-point learning as principled objectives, and (ii) a residual-based audit certificate that turns approximation error into a uniform bound on robust performance.

**Theorem 1** (Contraction and unique fixed point). *Assume $r$ is bounded and $\mathcal{W}$ is compact. Then $\mathcal{T}$ defined in (1) is a $\gamma$-contraction under $\|\cdot\|_\infty$ on $\mathcal{V}$. Hence, $\mathcal{T}$ admits a unique fixed point $V^\star$, and $V^\star$ equals the max–min value of the game.*

When instantiating $D_\Phi$ with KL divergence, we restrict $\mathcal{W}$ to an interior simplex to avoid boundary singularities; see Appendix C.5.

*Proof sketch.* Fix $V_1, V_2 \in \mathcal{V}$. For each $(s, w)$, the inner expression differs only through the term $\gamma V(s', w')$. Taking $\min_{w'}$ and $\max_a$ cannot increase pointwise differences, and expectation is linear. Therefore,

$$|(\mathcal{T}V_1)(s,w) - (\mathcal{T}V_2)(s,w)| \leq \gamma \, \|V_1 - V_2\|_\infty.$$

Taking supremum over $(s, w)$ yields the contraction. Existence and uniqueness of the fixed point follow from Banach's theorem, and the game interpretation follows from standard Bellman–Isaacs arguments. The complete proof is deferred to the appendix.

**Corollary 1** (Value iteration convergence). *For any $V_0 \in \mathcal{V}$, the iterates $V_{k+1} \leftarrow \mathcal{T}V_k$ satisfy $\|V_k - V^\star\|_\infty \leq \gamma^k \|V_0 - V^\star\|_\infty$, and thus converge to $V^\star$.*

The next result formalizes the exact solution concept for the grid implementation and motivates our certificate computation.

**Theorem 2** (Stationary saddle on a weight grid). *Suppose $\mathcal{S}$ and $\mathcal{A}$ are finite and the weight set is discretized to a finite grid $\mathcal{W}_G \subset \mathcal{W}$. Then the induced discounted zero-sum stochastic game on the augmented state space $\mathcal{S} \times \mathcal{W}_G$ admits a stationary saddle pair $(\pi^\star, \sigma^\star)$, and $V^\star$ satisfies*

$$V^\star(s, w) = J^{\pi^\star, \sigma^\star}(s, w) = \max_\pi \min_\sigma J^{\pi, \sigma}(s, w)$$
$$= \min_\sigma \max_\pi J^{\pi, \sigma}(s, w).$$

*Proof sketch.* With $\mathcal{W}_G$ finite, $(s, w)$ is a finite Markov game state. The one-step payoff is bounded and the discount factor satisfies $\gamma < 1$, so Shapley's theorem for discounted stochastic games applies to guarantee existence of stationary equilibria and a value function satisfying the Bellman–Isaacs fixed point equation on $\mathcal{S} \times \mathcal{W}_G$ (Shapley, 1953). The complete proof is deferred to the appendix.

*Continuous weights.* Beyond finite grids, under compactness and mild continuity assumptions on $r$, $P$, and $D_\Phi$, the same stationary Markov saddle and min–max equality hold for continuous compact $\mathcal{W}$; Appendix B.5 provides a proof based on Berge's maximum theorem and measurable selection.

**Theorem 3** (Bellman residual certificate). *For any $\widehat{V} \in \mathcal{V}$, define the Bellman residual $\varepsilon(\widehat{V}) \triangleq \left\| \mathcal{T}\widehat{V} - \widehat{V} \right\|_\infty$. Then*

$$\left\| \widehat{V} - V^\star \right\|_\infty \leq \frac{\varepsilon(\widehat{V})}{1 - \gamma}.$$

*Proof sketch.* Write $V^\star = \mathcal{T}V^\star$ and add/subtract $\mathcal{T}\widehat{V}$:

$$\left\| \widehat{V} - V^\star \right\|_\infty \leq \left\| \widehat{V} - \mathcal{T}\widehat{V} \right\|_\infty + \left\| \mathcal{T}\widehat{V} - \mathcal{T}V^\star \right\|_\infty$$
$$\leq \varepsilon(\widehat{V}) + \gamma \left\| \widehat{V} - V^\star \right\|_\infty.$$

where the second inequality uses Theorem 1. Rearranging yields the bound. The complete proof is deferred to the appendix.

**Theorem 4** (Residual-to-policy-performance certificate). *Let $\pi_{\widehat{V}}$ be any stationary greedy policy w.r.t. $\widehat{V}$: for all $(s, w)$,*

$$\pi_{\widehat{V}}(s, w) \in \arg\max_{a \in \mathcal{A}} \mathbb{E}_{s' \sim P(\cdot|s,a)} \Big[ \min_{w' \in \mathcal{W}} \big\{ \langle w', r(s, a, s') \rangle$$
$$+ \lambda D_\Phi(w' \mid w) + \gamma \widehat{V}(s', w') \big\} \Big].$$

*Let $V^{\pi_{\widehat{V}}}(s, w) \triangleq \min_\sigma J^{\pi_{\widehat{V}}, \sigma}(s, w)$ be the robust value of $\pi_{\widehat{V}}$. Then*

$$0 \leq V^\star - V^{\pi_{\widehat{V}}} \leq \frac{2\varepsilon(\widehat{V})}{1 - \gamma}.$$

*Proof sketch.* The greedy definition gives $\mathcal{T}\widehat{V} = \mathcal{T}^{\pi_{\widehat{V}}}\widehat{V}$ for the fixed-policy evaluation operator $\mathcal{T}^\pi$. Applying the same contraction argument as Theorem 3 yields $\|V^{\pi_{\widehat{V}}} - V^\star\|_\infty \leq 2\varepsilon(\widehat{V})/(1 - \gamma)$, and optimality gives the one-sided bound; see Appendix B.7.

**Proposition 1** ($\lambda$-monotonicity and limits). *Let $V_\lambda^\star$ denote the fixed point of $\mathcal{T}$ with switching-cost coefficient $\lambda$. If $\lambda_1 \leq \lambda_2$, then $V_{\lambda_1}^\star(s, w) \leq V_{\lambda_2}^\star(s, w)$ for all $(s, w)$. Moreover, as $\lambda \to \infty$, the optimal opponent's choice collapses to $w_{t+1} = w_t$, reducing (1) to a fixed-weight scalar MDP; as $\lambda \to 0$, the recursion approaches the unregularized step-wise inner minimization.*

*Proof sketch.* For fixed $V$, the inner objective $\langle w', r \rangle + \lambda D_\Phi(w' \mid w) + \gamma V(s', w')$ is pointwise nondecreasing in $\lambda$ for every $w'$, so its minimum over $w'$ is also nondecreasing; the outer $\max_a$ and expectation preserve the ordering. By monotone fixed-point arguments for contractions, the same ordering carries to $V_\lambda^\star$. For the limits, $D_\Phi(w' \mid w) \geq 0$ with equality at $w' = w$; as $\lambda$ grows, any deviation $w' \neq w$ incurs an increasingly dominant penalty, making $w' = w$ optimal for the minimizing opponent in the limit. The complete proof is deferred to the appendix.

*Remark* 1 (What the certificate buys). Theorem 3 is operational when $\varepsilon(\widehat{V})$ can be computed or bounded. On a finite grid $\mathcal{W}_G$, $\mathcal{T}\widehat{V}$ can be evaluated exactly and thus $\varepsilon(\widehat{V})$ becomes a computable certificate. In the deep regime, it becomes an empirical diagnostic: it quantifies how closely training approximates the Bellman–Isaacs fixed point.

## 5. Algorithms

All algorithms in this paper target the same operator (1). The difference between "tabular" and "deep" versions is purely computational: how we represent $V$ (or $Q$), and how we approximate the inner minimization over $w'$.

Operationally, a Bellman–Isaacs backup differs from a standard Bellman backup only through the additional inner optimization over the next weight. The key subtlety is that this inner step depends on the current critic via the continuation value, so the minimizing weight cannot be treated as a static component of the environment dynamics during learning. Our implementations therefore recompute the inner minimizer whenever forming a backup target, while keeping the surrounding fixed-point update identical to familiar dynamic programming (tabular) or DQN-style learning (deep).

### 5.1. Grid value iteration and certificates

When $\mathcal{W}$ is discretized to a finite grid $\mathcal{W}_G$, the game becomes a finite discounted stochastic game over $\mathcal{S} \times \mathcal{W}_G$. Value iteration is then a direct fixed-point method: repeatedly apply $\mathcal{T}$ until the change $\|V_{k+1} - V_k\|_\infty$ is small. Because $\mathcal{T}$ is a contraction, this yields both a convergent al-

gorithm (Corollary 1) and a certificate via Theorem 3. Algorithm 1 records the procedure. Beyond being an exact baseline, this grid solver provides an auditable certificate, which we use to validate approximate solvers.

---

**Algorithm 1** Grid value iteration with a Bellman residual certificate

1: Input: weight grid $\mathcal{W}_G$, discount $\gamma$, switching-cost $\lambda$, tolerance $\epsilon$.
2: Initialize $V_0(s, w) \leftarrow 0$ for all $(s, w) \in \mathcal{S} \times \mathcal{W}_G$.
3: **for** $k = 0, 1, 2, \ldots$ **do**
4:    $V_{k+1} \leftarrow \mathcal{T}V_k$ using Eq. (1) with $\min_{w' \in \mathcal{W}_G}$.
5:    **if** $\|V_{k+1} - V_k\|_\infty \leq \epsilon(1 - \gamma)$ **then**
6:       **break**
7:    **end if**
8: **end for**
9: Certificate: Residual $\leftarrow \|\mathcal{T}V_{k+1} - V_{k+1}\|_\infty$ and error bound Residual/$(1 - \gamma)$.
10: Output: $V_{k+1}$ and certificate.

---

### 5.2. Deep fixed-point iteration with inner Bregman updates

In continuous $\mathcal{W}$ and with function approximation, the computational burden concentrates in the inner minimization. Given $(s', w, r)$ and a continuation value $V$, define the opponent's one-step best response $w^\star \equiv w^\star(s', w, r; V)$ as any minimizer:

$$w^\star \in \arg\min_{w' \in \mathcal{W}} \{\langle w', r \rangle + \lambda D_\Phi(w' \mid w) + \gamma V(s', w')\}.$$

We interpret this step as a Bregman-proximal (mirror) update induced by $\Phi$: for KL it reduces to an exponentiated-gradient update, and for $\ell_2$ it becomes a projected gradient step (Appendix C.5). When $\mathcal{W}$ is a small grid (e.g., $d \in \{2, 3\}$), we can also solve the minimization exactly by enumerating $w' \in \mathcal{W}_G$. For training stability on a finite grid, we optionally replace the inner minimum with a softmin surrogate; Appendix C.4 gives the induced approximation bound. Unlike storing $w_{t+1}$ in replay, we recompute $w^\star$ on-the-fly using the current target critic so that learning continues to target (1).

Algorithm 2 summarizes the deep fixed-point iteration (DQN-style) for an action-value function $Q_\theta(s, w, a)$ on the augmented state.

## 6. Evaluation Protocol

Robustness claims in this paper are about performance under a step-wise opponent. Evaluating a learned policy against such an opponent is not a routine "test-time" choice: different protocols correspond to different opponent strengths, and weak protocols can systematically overestimate robustness. Unless stated otherwise, all reported WRR/DRIFT/GAP values (defined below) are computed under BR-$K$: we train $K$ independently initialized stepwise best-response adversaries and report the metrics for the adversary that achieves the lowest WRR.

---

**Algorithm 2** Deep Bellman–Isaacs Q-iteration with on-the-fly inner minimization

1: Initialize $Q_\theta$, target $Q_{\bar{\theta}}$, replay buffer $\mathcal{D}$.
2: **for** env steps $t = 0, 1, 2, \ldots$ **do**
3:    Observe $(s_t, w_t)$.
4:    Select $a_t$ by $\epsilon$-greedy on $Q_\theta(s_t, w_t, \cdot)$.
5:    Execute $a_t$, observe $s_{t+1}$ and vector reward $r_t$.
6:    Compute $w_{t+1}$ by approximately solving the inner minimization in (1) (using INNERMIN).
7:       INNERMIN: $J$ mirror/prox steps on $w$ (KL: exponentiated-gradient; $\ell_2$: projection) with gradients through $Q_{\bar{\theta}}$.
8:    Store $(s_t, w_t, a_t, r_t, s_{t+1}, d_t)$ in $\mathcal{D}$, where $d_t \in \{0, 1\}$ indicates termination.
9:    Sample minibatch from $\mathcal{D}$.
10:    **for** each sample $(s, w, a, r, s_+, d)$ **do**
11:       Compute $w^\star \leftarrow$ INNERMIN$(s_+, w, r; Q_{\bar{\theta}})$ (inner minimization in (1)).
12:       Set target $y \leftarrow \langle w^\star, r \rangle + \lambda D_\Phi(w^\star \mid w) + \gamma(1 - d) \max_{a'} Q_{\bar{\theta}}(s_+, w^\star, a')$.
13:    **end for**
14:    Update $\theta$ by a gradient step on $\left(Q_\theta(s, w, a) - y\right)^2$.
15:    Periodically update $\bar{\theta} \leftarrow \theta$.
16: **end for**

---

### 6.1. Metrics

For a rollout $(s_t, a_t, w_t)_{t \geq 0}$, define the weighted reward return (WRR), the discounted drift (DRIFT), and the game return (GAME) optimized in the zero-sum game:

$$\text{WRR} := \sum_{t \geq 0} \gamma^t \langle w_{t+1}, r_t \rangle,$$

$$\text{DRIFT} := \sum_{t \geq 0} \gamma^t D_\Phi(w_{t+1} \mid w_t),$$

$$\text{GAME} := \sum_{t \geq 0} \gamma^t \left(\langle w_{t+1}, r_t \rangle + \lambda D_\Phi(w_{t+1} \mid w_t)\right).$$

We train both the agent and the opponent to optimize GAME, but report (WRR, DRIFT) separately to make the return–drift tradeoff explicit. For $d = 2$, we additionally report diagnostics based on the discounted vector return $R := \sum_{t \geq 0} \gamma^t r_t$, namely $\min(R_1, R_2)$ and GAP $:= \max(R_1, R_2) - \min(R_1, R_2)$.

We use BR-$K$ throughout to evaluate a fixed policy against strong step-wise opponents; Algorithm 3 gives the protocol.

**Algorithm 3** BR-$K$ evaluation for a fixed policy $\pi$

---

1: Input: frozen policy $\pi$, weight grid $\mathcal{W}_G$, switching-cost $\lambda$, restarts $K$, BR budget $N$.
2: **for** $k = 1, 2, \ldots, K$ **do**
3:     Initialize adversary $Q_{\psi_k}^{\mathrm{adv}}$ and a replay buffer $\mathcal{B}$.
4:     **for** BR steps $n = 1, 2, \ldots, N$ **do**
5:         Roll out one step with $\pi$ to obtain $(s_t, w_t, a_t, s_{t+1}, r_t)$.
6:         Set adversary observation $z_t = (s_{t+1}, w_t, r_t)$ (or $(s_{t+1}, w_t)$).
7:         Choose $w_{t+1} \in \mathcal{W}_G$ by $\epsilon$-greedy on $Q_{\psi_k}^{\mathrm{adv}}(z_t, \cdot)$.
8:         Compute stage payoff $g_t \leftarrow \langle w_{t+1}, r_t \rangle + \lambda D_\Phi(w_{t+1} \mid w_t)$.
9:         Set adversary reward $\rho_t \leftarrow -g_t$; store transition and update $\psi_k$ (DQN).
10:     **end for**
11:     Evaluate $\pi$ vs greedy $\sigma_{\psi_k}$ to estimate $(\widehat{\mathrm{WRR}}_k, \widehat{\mathrm{DRIFT}}_k)$.
12: **end for**
13: Output: report $\min_k \widehat{\mathrm{WRR}}_k$ and the corresponding DRIFT / auxiliary metrics.

---

### 6.2. Best-response adversary and BR-$K$

As a weak sanity-check baseline, we also consider a myopic reward-only opponent that ignores the continuation value and chooses

$$w_{t+1} \in \operatorname*{arg\,min}_{w' \in \mathcal{W}} \left\{ \langle w', r_t \rangle + \lambda D_\Phi(w' \mid w_t) \right\}.$$

This opponent is useful for debugging but cannot capture long-horizon vulnerabilities where optimal adversarial choices depend on future states.

To approximate a strong step-wise opponent for a fixed policy $\pi$, we train an adversary with its own function approximator (a DQN in our deep experiments) to minimize the full discounted objective under the game information structure. Concretely, the adversary observes $(s_{t+1}, w_t, r_t)$ (or $(s_{t+1}, w_t)$ in weak-ablation studies) and chooses the next weight $w_{t+1}$. In our deep experiments the adversary's action space is the same discrete weight grid $\mathcal{W}_G$ used for evaluation, so choosing $w_{t+1}$ is a discrete decision.

Training a best-response adversary is itself a nonconvex RL problem, and single runs can yield weak opponents. We therefore use BR-$K$: repeat BR training $K$ times with different random seeds and report the minimum WRR across the $K$ trained opponents.

### 7. Experiments

Our empirical goal is to test whether solving the time-consistent Bellman–Isaacs recursion translates into improved performance under a strong step-wise opponent, and

to stress-test these conclusions under different evaluation assumptions (opponent information, initial weight, and divergence geometry).

Our main figures evaluate on two MO-Gymnasium MountainCar variants (Timespeed and Timemove) and DeepSeaTreasure (DST). These are two-objective problems with weights on the 2D simplex. We compare against common preference-conditioned MORL baselines: conditioned networks (CN-DQN) (Abels et al., 2019), envelope methods (Yang et al., 2019), and Pareto-conditioned networks (PCN) (Reymond et al., 2022), as well as a fixed-weight DQN trained at a single scalarization. We do not include *static* max–min robust MORL methods (e.g., Park et al. (2024)) since they optimize robustness to an unknown but fixed weight chosen once per episode, whereas our focus is a *step-wise* (state-reactive) weight adversary. All methods are evaluated under the same step-wise adversary via BR-$K$ (Section 6).

#### 7.1. Robustness under a strong step-wise opponent

Figure 2 summarizes BR-$K$ evaluation ($K = 3$ restarts, best-response training budget 200k steps) at switching-cost strength $\lambda = 1.0$ and initial weight $w_0 = (0.5, 0.5)$. Across Timespeed, Timemove, and DeepSeaTreasure, our method achieves the highest WRR while keeping the induced DRIFT small. On $d = 2$ tasks, we additionally report GAP derived from the vector return; the same pattern holds: higher WRR does not come from large DRIFT, but from learning trajectories that remain good under adversarial reweighting.

Appendix F.2 reports the exact WRR and DRIFT values for the MountainCar panels of Figure 2 (under BR-$K$). Appendix F.3 further reports higher-dimensional stress tests on 4-objective MO-Lunar-Lander and 6-objective Fruit-Tree, together with a continuous TD3 best-response adversary on DeepSeaTreasure and BR-$K$ sensitivity checks.

*Implementation details.* Unless explicitly stated in the appendix stress tests, the main deep experiments use two-objective tasks ($d = 2$) and evaluate robustness over a discrete weight grid on the simplex. Unless otherwise stated, BR-$K$ uses $K = 3$ independently trained best-response adversaries with a fixed training budget (200k BR steps) and we report the adversary that achieves the lowest mean WRR against the frozen agent. Complete environment, network, and hyperparameter details are provided in Appendix E.

### 8. Discussion and Limitations

*Information structure.* Our main robustness definition and BR-K protocol assume the strong opponent that observes $s_{t+1}$ (and, in our strongest setting, also the realized reward $r_t$) before choosing $w_{t+1}$. This captures reactive preference shifts triggered by outcomes and yields a Markov recursion,

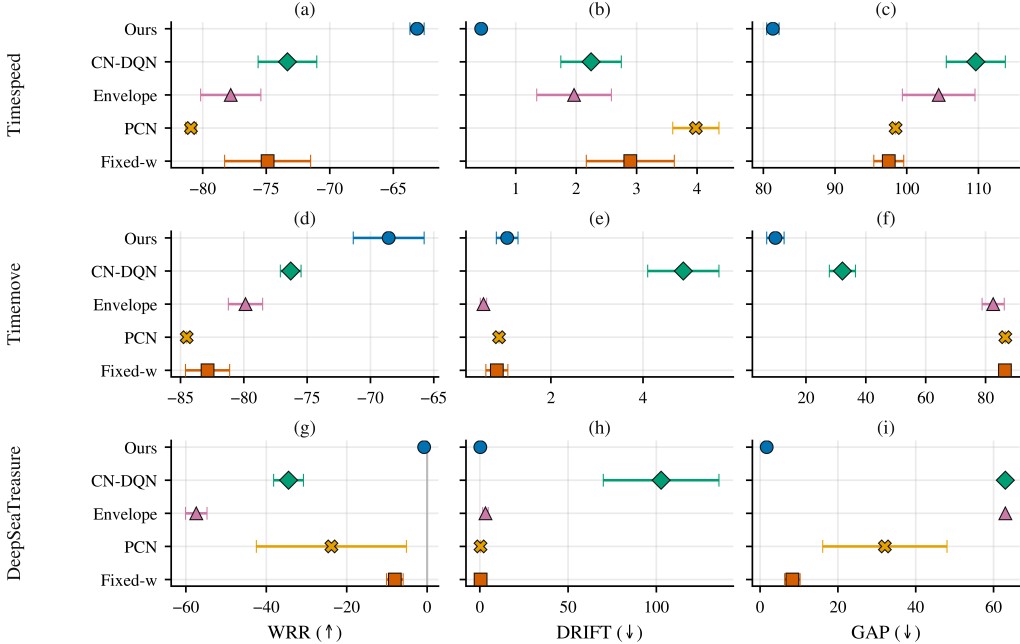

*Figure 2.* Aggregate BR-$K$ evaluation on Timespeed, Timemove, and DeepSeaTreasure at $\lambda = 1.0$ and $w_0 = (0.5, 0.5)$. Columns report (left) WRR (higher is better), (middle) DRIFT (lower is better), and (right) GAP computed from the discounted vector return $R$ (lower is better). For each method, we freeze an agent checkpoint, train $K = 3$ independent step-wise best-response adversaries, and report the adversary that achieves the lowest WRR (with its associated DRIFT / GAP). Error bars show standard error across independent agent seeds.

but it is a modeling choice: in some domains, preference shifts may depend on latent context or change more slowly than the environment. A move-first preference opponent, which must choose $w_{t+1}$ before observing the realized transition outcome, would induce a different and weaker information structure and generally a different Bellman operator. Our results should therefore be read as characterizing the after-outcome, reactive-preference game; move-first robustness is a complementary variant, not a special case claimed here. We report ablations with weaker opponent observations in Appendix F.4, and extending the framework to partial observability or explicit switching constraints beyond a Bregman penalty is an important direction.

*Computation and scalability.* The step-wise inner minimization adds per-step overhead: on a weight grid it scales with $|\mathcal{W}_G|$ by enumeration, and in continuous $\mathcal{W}$ we approximate it with a small number of mirror/prox steps. BR-K evaluation further multiplies cost by training $K$ independent adversaries. While BR-K reduces evaluation variance and avoids under-trained opponents, practical use may require more sample-efficient adversary learning or adaptive early stopping. Although the additional 4D/6D stress tests in Appendix F.3 suggest that the method is not tied to two objectives, scaling to much higher-dimensional objective vectors still raises challenges, since both the inner optimization and the representation of $Q(s, w, a)$ become harder and may require richer mirror maps.

*Approximation gap.* Our theory concerns the exact Bellman–Isaacs operator. Deep implementations rely on approximation choices (e.g., finite mirror steps, softmin smoothing, function approximation), so the learned policy may solve an approximate game. The Bellman residual remains a useful diagnostic, but formal residual-to-performance guarantees under function approximation for discounted stochastic games remain open.

*Summary.* We argue that robust MORL under state-reactive preference shifts is naturally modeled by placing the opponent's weight choice *inside* the Bellman backup, yielding a time-consistent Bellman–Isaacs fixed point. Our contraction and Bellman-residual certificate make this objective well-defined and auditable, our grid and deep solvers provide practical approximations to the same operator, and BR-K offers a method-agnostic protocol for evaluating against strong step-wise opponents. Across MO-Gymnasium benchmarks, this combination improves WRR while keeping DRIFT controllable via $\lambda$.

## Impact Statement

This paper presents work whose goal is to advance the field of machine learning. There are many potential societal consequences of our work, none of which we feel must be specifically highlighted here.

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

# Appendix Roadmap

The appendix is organized to make it easy to locate proofs, technical lemmas, and experimental details. Table 1 links each appendix section to the corresponding claims and artifacts.

*Table 1.* Appendix roadmap (what to read and why).

| Appendix | Contents | Used for / connects to |
|---|---|---|
| A | Time-consistency viewpoint and a one-step preference–risk operator. | Conceptual link to the nested Bellman–Isaacs recursion (Sec. 3). |
| B | Full proofs of the contraction / fixed-point results, stationary saddle on a grid, and residual certificates. | Completes Main theoretical guarantees (Thms. 1–4). |
| C | Solver-facing technical results (closed-form inner steps; softmin approximation). | Supports algorithmic details in Sec. 5 and the empirical protocol described in Appendix E. |
| D | Additional analyses (e.g., inner-vs-outer separation; tabular remarks). | Context for modeling/design choices and limitations discussion. |
| E | Experimental setup, metrics, and BR/BR-$K$ evaluation protocol. | Reproducibility of the empirical pipeline and evaluation assumptions. |
| F | Additional experiments and diagnostics (initial-weight sensitivity; numeric tables; additional stress tests; ablations). | Supports Sec. 7 claims and helps interpret Fig. 2. |

**Appendix guide.** For convenience, we begin with two lookup tables: one for notation (Table 2) and one summarizing the main guarantees and proof locations (Table 3). The numerical counterpart of Fig. 2 is reported in Appendix F.2.

*Table 2.* Key notation (abridged).

| Symbol | Meaning |
|---|---|
| $\mathcal{S}, \mathcal{A}$ | State and action spaces. |
| $w \in \mathcal{W} \subset \mathbb{R}_+^d$ | Preference / scalarization weight (often $\|w\|_1 = 1$). |
| $\Phi$ | Convex regularizer generating a divergence $D_\Phi(\cdot \mid \cdot)$. |
| $\lambda$ | Strength of the divergence/"drift" penalty. |
| $J^{\pi,\sigma}(s, w)$ | Discounted utility under agent policy $\pi$ and adversary $\sigma$. |
| $V^\star$ | Optimal robust value: $V^\star = \max_\pi \min_\sigma J^{\pi,\sigma}$. |
| $\mathcal{T}$ | Time-consistent Bellman–Isaacs operator (Sec. 3; Appendix A). |
| $\varepsilon(\widehat{V})$ | Bellman residual: $\|\mathcal{T}\widehat{V} - \widehat{V}\|_\infty$. |
| WRR | Weighted reward return $\sum_t \gamma^t \langle w_{t+1}, r_t \rangle$. |
| DRIFT | Discounted preference drift $\sum_t \gamma^t D_\Phi(w_{t+1} \mid w_t)$. |
| GAP | Gap from the discounted vector return $R$: $\max_i R_i - \min_i R_i$ (for the main $d = 2$ plots this reduces to $\max(R_1, R_2) - \min(R_1, R_2)$). |
| BR / BR-$K$ | Best-response adversary; $K$ independent restarts; we report the restart with lowest WRR. |

*Table 3.* Main results overview.

| Result / task | Guarantee | Where |
|---|---|---|
| Contraction / fixed point | $\|\mathcal{T}V - \mathcal{T}U\|_\infty \leq \gamma \|V - U\|_\infty$; hence a unique fixed point $V^\star$ exists and value iteration satisfies $\|V_k - V^\star\|_\infty \to 0$. | Thm. 1 |
| Stationary saddle (grid) | For finite $\mathcal{S}, \mathcal{A}$ and finite $\mathcal{W}_G$, the induced discounted game admits a stationary saddle pair $(\pi^\star, \sigma^\star)$ and $V^\star = \max_\pi \min_\sigma J^{\pi,\sigma}$ $= \min_\sigma \max_\pi J^{\pi,\sigma}$. | Thm. 2 |
| Stationary saddle (continuous) | Under compact $\mathcal{W}$ and mild continuity/compactness assumptions, the same Markov saddle and min–max equality hold for continuous weights. | App. B.5 |
| Bellman residual certificate | For any $\widehat{V}$, letting $\varepsilon(\widehat{V}) = \|\mathcal{T}\widehat{V} - \widehat{V}\|_\infty$, we have $\|\widehat{V} - V^\star\|_\infty \leq \varepsilon(\widehat{V})/(1-\gamma)$. | Thm. 3 |
| Residual-to-policy performance | Any stationary greedy policy $\pi_{\widehat{V}}$ satisfies $0 \leq V^\star - V^{\pi_{\widehat{V}}} \leq 2\varepsilon(\widehat{V})/(1-\gamma)$. | Thm. 4 |
| $\lambda$-limits and monotonicity | The optimal robust value $V_\lambda^\star$ is nondecreasing in $\lambda$; $\lambda \to 0$ removes the switching-cost regularization, yielding the unregularized step-wise (inner) adversary; and $\lambda \to \infty$ recovers fixed-weight behavior (no drift). | Prop. 1 |

## A. Time consistency and a one-step preference–risk operator

A distinguishing feature of our formulation is that the preference adversary acts *inside* each one-step Bellman backup. This section makes explicit that the Bellman–Isaacs recursion can be written as a nested composition of a one-step "preference–risk" operator, placing the objective in the standard time-consistent dynamic-evaluation form studied in risk-sensitive control and dynamic risk measures (Ruszczyński, 2010).

### A.1. A one-step operator on continuation values

Fix a realized tuple $(s', w, r)$ and let $X : \mathcal{W} \to \mathbb{R}$ be any bounded continuation slice. Define

$$\rho_{s',w,r}(X) \triangleq \min_{w' \in \mathcal{W}} \left\{ \langle w', r \rangle + \lambda D_\Phi(w' \mid w) + \gamma X(w') \right\}. \tag{2}$$

Then the Bellman–Isaacs operator in Eq. (1) can be written compactly as

$$(\mathcal{T}V)(s,w) = \max_{a \in \mathcal{A}} \mathbb{E}_{s' \sim P(\cdot|s,a)} \left[ \rho_{s',w,r(s,a,s')}(V(s',\cdot)) \right].$$

**Lemma 1** (Monotonicity and discounted translation). *For any bounded $X, Y : \mathcal{W} \to \mathbb{R}$ and any constant $c \in \mathbb{R}$, the one-step operator (2) satisfies: (i) monotonicity: if $X \leq Y$ pointwise then $\rho_{s',w,r}(X) \leq \rho_{s',w,r}(Y)$; (ii) discounted translation: $\rho_{s',w,r}(X + c) = \rho_{s',w,r}(X) + \gamma c$; and (iii) Lipschitz continuity: $|\rho_{s',w,r}(X) - \rho_{s',w,r}(Y)| \leq \gamma \|X - Y\|_\infty$.*

*Proof.* All statements are direct from the definition. For (i), if $X \leq Y$ then the inner objective for $X$ is pointwise no larger than that for $Y$, so their minima satisfy the same ordering. For (ii), adding a constant $c$ to $X$ adds $\gamma c$ to every candidate in the minimization, hence to the minimum. For (iii), apply Lemma 2 to the minimization and use $\sup_{w'} |\gamma X(w') - \gamma Y(w')| = \gamma \|X - Y\|_\infty$. $\square$

### A.2. Nested evaluation and time consistency

Consider a finite horizon $H$ and define $V_H(s,w) \equiv 0$. For $t = H-1, \ldots, 0$, define recursively

$$V_t(s,w) = \max_{a \in \mathcal{A}} \mathbb{E}_{s' \sim P(\cdot|s,a)} \left[ \rho_{s',w,r(s,a,s')}(V_{t+1}(s',\cdot)) \right]. \tag{3}$$

This is exactly the dynamic-programming recursion obtained by backward induction in the step-wise preference game. Lemma 1 implies that each stage mapping in (3) is a monotone one-step conditional evaluation with discounted translation. As a consequence, the induced objective is *time-consistent*: the optimal continuation from time $t$ depends only on the current augmented state $(s_t, w_t)$ and coincides with the continuation prescribed by the recursion at earlier times (the principle of optimality). In the infinite-horizon discounted case, the same nesting appears in the fixed point of $\mathcal{T}$, so $V^\star$ inherits this time-consistent dynamic-programming structure.

# B. Proofs of Main Results

This section provides full proofs for the contraction and fixed-point characterization (Theorem 1 and Corollary 1), the existence of a stationary saddle on a weight grid (Theorem 2), and the Bellman residual certificate (Theorem 3), supporting the theoretical claims in Sec. 4.

## B.1. Auxiliary inequalities for max and min

**Lemma 2.** *Let $\mathcal{X}$ be a nonempty set and let $f, g : \mathcal{X} \to \mathbb{R}$. Then*

$$\Big| \min_{x \in \mathcal{X}} f(x) - \min_{x \in \mathcal{X}} g(x) \Big| \leq \sup_{x \in \mathcal{X}} |f(x) - g(x)|,$$

$$\Big| \max_{x \in \mathcal{X}} f(x) - \max_{x \in \mathcal{X}} g(x) \Big| \leq \sup_{x \in \mathcal{X}} |f(x) - g(x)|.$$

*Proof.* We prove the min inequality; the max inequality is analogous. Let $\Delta := \sup_{x \in \mathcal{X}} |f(x) - g(x)|$. For any $x \in \mathcal{X}$, we have $f(x) \leq g(x) + \Delta$. Taking $\min_{x \in \mathcal{X}}$ on both sides yields $\min_x f(x) \leq \min_x g(x) + \Delta$. Swapping $f$ and $g$ gives $\min_x g(x) \leq \min_x f(x) + \Delta$. Combining the two inequalities proves the claim. $\square$

## B.2. Proof of Theorem 1

*Proof.* We show that $\mathcal{T}$ maps bounded functions to bounded functions and is a $\gamma$-contraction under $\| \cdot \|_\infty$. Banach's fixed-point theorem then yields the unique fixed point and geometric convergence of value iteration.

*Well-definedness.* Let $\mathcal{B}$ be the space of bounded functions $V : \mathcal{S} \times \mathcal{W} \to \mathbb{R}$ with norm $\|V\|_\infty := \sup_{(s,w)} |V(s,w)|$. Assume the vector reward is uniformly bounded in the sense that $\|r(s,a,s')\|_\infty \leq R_{\max}$ for all $(s,a,s')$ and that the divergence is bounded on $\mathcal{W}$, i.e., $D_\Phi(w' \mid w) \leq D_{\max}$ for all $(w,w') \in \mathcal{W}^2$. Since $w' \in \mathcal{W} \subseteq \Delta^d$, we have $|\langle w', r(s,a,s') \rangle| \leq R_{\max}$. Therefore, for any bounded $V \in \mathcal{B}$ and any tuple $(s,w,a,s',w')$, writing $r := r(s,a,s')$,

$$\big| \langle w', r \rangle + \lambda D_\Phi(w' \mid w) + \gamma V(s', w') \big| \leq R_{\max} + \lambda D_{\max} + \gamma \|V\|_\infty.$$

Taking $\min_{w'}$, expectation over $s'$, and $\max_a$ preserves boundedness, so $(\mathcal{T}V)(\cdot, \cdot)$ is bounded and $\mathcal{T} : \mathcal{B} \to \mathcal{B}$.

*Contraction.* Fix $V, U \in \mathcal{B}$ and $(s,w)$. For each action $a \in \mathcal{A}$, define

$$F_V(a) := \mathbb{E}_{s' \sim P(\cdot|s,a)} \Big[ \min_{w' \in \mathcal{W}} \big( c(s,w,a,s',w') + \gamma V(s',w') \big) \Big].$$

where $c(s,w,a,s',w') := \langle w', r(s,a,s') \rangle + \lambda D_\Phi(w' \mid w)$ does not depend on $V$. Then $(\mathcal{T}V)(s,w) = \max_{a \in \mathcal{A}} F_V(a)$ and similarly for $U$. By Lemma 2 (applied to $\max_a$),

$$\big| (\mathcal{T}V)(s,w) - (\mathcal{T}U)(s,w) \big| \leq \max_{a \in \mathcal{A}} |F_V(a) - F_U(a)|.$$

Fix any action $a$. Using Lemma 2 again for the inner $\min_{w'}$ yields

$$|F_V(a) - F_U(a)| \leq \mathbb{E}_{s'} \Big[ \sup_{w' \in \mathcal{W}} \gamma |V(s',w') - U(s',w')| \Big]$$

$$\leq \gamma \|V - U\|_\infty.$$

Taking a maximum over $a$ gives $|(\mathcal{T}V)(s,w) - (\mathcal{T}U)(s,w)| \leq \gamma \|V - U\|_\infty$. Finally, taking $\sup_{(s,w)}$ proves $\|\mathcal{T}V - \mathcal{T}U\|_\infty \leq \gamma \|V - U\|_\infty$.

*Unique fixed point.* Since $\mathcal{T}$ is a $\gamma$-contraction on the complete metric space $(\mathcal{B}, \| \cdot \|_\infty)$, Banach's fixed-point theorem implies that $\mathcal{T}$ admits a unique fixed point $V^\star$ and that value iteration $V_{k+1} = \mathcal{T}V_k$ converges geometrically to $V^\star$. $\square$

## B.3. Proof of Corollary 1

*Proof.* Let $V^\star$ be the unique fixed point of $\mathcal{T}$. By Theorem 1, $\mathcal{T}$ is a $\gamma$-contraction under $\| \cdot \|_\infty$. Therefore,

$$\|V_{k+1} - V^\star\|_\infty = \|\mathcal{T}V_k - \mathcal{T}V^\star\|_\infty \leq \gamma \|V_k - V^\star\|_\infty.$$

Iterating the inequality yields $\|V_k - V^\star\|_\infty \leq \gamma^k \|V_0 - V^\star\|_\infty$. $\square$

## B.4. Proof of Theorem 2

*Proof.* We explicitly construct stationary policies $(\pi^\star, \mu^\star)$ from the Bellman–Isaacs fixed point and verify the saddle inequalities. Throughout this proof we work on the finite augmented state space $\mathcal{X} := \mathcal{S} \times \mathcal{W}_G$.

*Fixed-policy evaluation.* Fix any stationary Markov agent policy $\pi(\cdot \mid s, w)$ and any stationary Markov adversary policy $\mu(\cdot \mid s', w, a)$ (the adversary acts after observing $s'$ and may depend on the realized action $a$ through the observed vector reward). Define the policy-evaluation operator $\mathcal{T}^{\pi,\mu}$ by

$$(\mathcal{T}^{\pi,\mu}V)(s,w) := \mathbb{E}\big[\langle w', r(s,a,s')\rangle + \lambda D_\Phi(w' \mid w) + \gamma V(s', w')\big].$$

where $a \sim \pi(\cdot \mid s, w)$, $s' \sim P(\cdot \mid s, a)$, and $w' \sim \mu(\cdot \mid s', w, a)$. This is a $\gamma$-contraction under $\|\cdot\|_\infty$ by the same calculation as in Theorem 1 (linearity of expectation replaces $\max / \min$). Hence $\mathcal{T}^{\pi,\mu}$ admits a unique fixed point, denoted $V^{\pi,\mu}$, which equals the discounted return of the game under $(\pi, \mu)$.

*Greedy policies induced by $V^\star$.* Let $V^\star$ be the unique fixed point of $\mathcal{T}$ on $\mathcal{X}$. Because $\mathcal{A}$ and $\mathcal{W}_G$ are finite, maximizers and minimizers exist. For each $(s, w) \in \mathcal{X}$, pick

$$\pi^\star(s,w) \in \arg\max_{a\in\mathcal{A}} \mathbb{E}_{s'\sim P(\cdot|s,a)}\Big[\min_{w'\in\mathcal{W}_G} \big\{\langle w', r(s,a,s')\rangle + \lambda D_\Phi(w' \mid w) + \gamma V^\star(s', w')\big\}\Big].$$

For each tuple $(s, w, a, s')$, pick

$$\mu^\star(s',w,a) \in \arg\min_{w'\in\mathcal{W}_G} \big\{\langle w', r(s,a,s')\rangle + \lambda D_\Phi(w' \mid w) + \gamma V^\star(s', w')\big\}.$$

We view $(\pi^\star, \mu^\star)$ as deterministic stationary policies. By construction,

$$\mathcal{T}V^\star = \mathcal{T}^{\pi^\star,\mu^\star} V^\star. \tag{4}$$

Since $V^\star$ is a fixed point of $\mathcal{T}$, (4) implies $V^\star$ is also a fixed point of $\mathcal{T}^{\pi^\star,\mu^\star}$. By uniqueness of the fixed point of $\mathcal{T}^{\pi^\star,\mu^\star}$, we conclude

$$V^{\pi^\star,\mu^\star} = V^\star. \tag{5}$$

*A comparison lemma.* We use the following standard fact: if $F$ is a monotone $\gamma$-contraction with fixed point $V_F$, then $V \geq FV$ implies $V \geq V_F$, and $V \leq FV$ implies $V \leq V_F$. Indeed, if $V \geq FV$, then monotonicity gives $V \geq FV \geq F^2V \geq \cdots$, and the sequence converges to $V_F$ by contraction. The other direction is analogous.

*Saddle inequalities.* Fix any agent policy $\pi$. Since $\mathcal{T}$ applies a pointwise maximization over actions, for every $(s, w)$ we have

$$V^\star(s,w) = (\mathcal{T}V^\star)(s,w)$$
$$\geq \mathbb{E}_{a\sim\pi(\cdot|s,w)}\Big[\mathbb{E}_{s'\sim P(\cdot|s,a)}\big[\min_{w'\in\mathcal{W}_G} \{\langle w', r(s,a,s')\rangle + \lambda D_\Phi(w' \mid w) + \gamma V^\star(s', w')\}\big]\Big].$$

Replacing the inner minimum by evaluation at the specific minimizer $\mu^\star(s', w, a)$ yields $V^\star \geq \mathcal{T}^{\pi,\mu^\star} V^\star$ pointwise. By the comparison lemma, $V^\star \geq V^{\pi,\mu^\star}$. Combining with (5) gives $V^{\pi,\mu^\star} \leq V^{\pi^\star,\mu^\star}$.

Similarly, fix any adversary policy $\mu$. For every $(s, w)$, denote $a^\star := \pi^\star(s, w)$.

$$V^\star(s,w) = (\mathcal{T}V^\star)(s,w)$$
$$= \mathbb{E}_{s'\sim P(\cdot|s,a^\star)}\Big[\min_{w'\in\mathcal{W}_G} \big\{\langle w', r(s,a^\star,s')\rangle + \lambda D_\Phi(w' \mid w) + \gamma V^\star(s', w')\big\}\Big]$$
$$\leq \mathbb{E}_{s', \tilde{w}}\Big[\langle \tilde{w}, r(s,a^\star,s')\rangle + \lambda D_\Phi(\tilde{w} \mid w) + \gamma V^\star(s', \tilde{w})\Big],$$

where $\tilde{w} \sim \mu(\cdot \mid s', w, a^\star)$ and we used that $\min_{w'} \leq$ evaluation at any feasible $\tilde{w}$. This shows $V^\star \leq \mathcal{T}^{\pi^\star,\mu} V^\star$. By the comparison lemma, $V^\star \leq V^{\pi^\star,\mu}$. Using (5) again yields $V^{\pi^\star,\mu^\star} \leq V^{\pi^\star,\mu}$.

We have shown that for all stationary policies $(\pi, \mu)$,

$$V^{\pi,\mu^\star} \leq V^{\pi^\star,\mu^\star} \leq V^{\pi^\star,\mu},$$

i.e., $(\pi^\star, \mu^\star)$ is a stationary saddle pair. $\qquad\square$

## B.5. Continuous weights: stationary Markov equilibrium and min–max exchange

Theorem 2 assumes a finite weight grid in order to invoke Shapley's theorem directly and avoid measurability issues. In many settings the underlying preference set $\mathcal{W}$ is naturally continuous (e.g., a simplex), and the same equilibrium picture holds under standard compactness/continuity assumptions. We record a self-contained extension here.

**Theorem 5** (Stationary equilibrium for continuous $\mathcal{W}$). *Assume $\mathcal{S}$ and $\mathcal{A}$ are finite and $\mathcal{W} \subset \mathbb{R}^d$ is a nonempty compact set. Assume $r(s, a, s')$ is bounded and $D_\Phi(\cdot \mid \cdot)$ is continuous on $\mathcal{W} \times \mathcal{W}$. Consider the Bellman–Isaacs operator (1) acting on the space of functions $V : \mathcal{S} \times \mathcal{W} \to \mathbb{R}$ that are continuous in $w$ for every fixed $s$. Then the following hold: (i) $\mathcal{T}$ maps this space into itself and is a $\gamma$-contraction under $\| \cdot \|_\infty$, hence it admits a unique fixed point $V^\star$ that is continuous in $w$. (ii) There exist measurable stationary Markov policies $(\pi^\star, \mu^\star)$ (agent and preference adversary) such that $V^\star = J^{\pi^\star, \mu^\star}$ and the saddle equalities*

$$V^\star(s, w) = \max_\pi \min_\mu J^{\pi, \mu}(s, w) = \min_\mu \max_\pi J^{\pi, \mu}(s, w)$$

*hold for all $(s, w)$.*

*Proof.* We follow the same greedy-policy construction as in the finite-grid proof, with compactness ensuring that the one-step $\min$ is attained and measurability ensured by a standard selection argument.

*(i) Continuity preservation.* Fix $V$ continuous in $w$ (for each $s$). For any fixed tuple $(s, w, a, s')$, the inner objective $w' \mapsto \langle w', r(s, a, s') \rangle + \lambda D_\Phi(w' \mid w) + \gamma V(s', w')$ is continuous on compact $\mathcal{W}$, hence its minimum is attained. Moreover, by Berge's maximum theorem, the value of this parametric minimization is continuous in $w$. Taking expectation over the finite next-state set $\mathcal{S}$ and the maximum over the finite action set $\mathcal{A}$ preserves continuity, hence $(\mathcal{T}V)(s, \cdot)$ is continuous for each $s$. The $\gamma$-contraction property is unchanged from Theorem 1, so the unique fixed point $V^\star$ exists and is continuous.

*(ii) Greedy selectors and saddle.* For each $(s, w)$, define the action-value

$$Q^\star(s, w, a) := \mathbb{E}_{s' \sim P(\cdot \mid s, a)} \Big[ \min_{w' \in \mathcal{W}} \{ \langle w', r(s, a, s') \rangle + \lambda D_\Phi(w' \mid w) + \gamma V^\star(s', w') \} \Big].$$

By the continuity argument above, $Q^\star(s, \cdot, a)$ is continuous for each $a$, hence the correspondence $\arg\max_{a \in \mathcal{A}} Q^\star(s, w, a)$ is nonempty. Since $\mathcal{A}$ is finite, we can pick a measurable maximizer $\pi^\star(s, w)$ by any fixed tie-breaking rule. Similarly, for each realized $(s, w, a, s')$, the correspondence of minimizers

$$\arg\min_{w' \in \mathcal{W}} \{ \langle w', r(s, a, s') \rangle + \lambda D_\Phi(w' \mid w) + \gamma V^\star(s', w') \}$$

is nonempty and compact. By a measurable selection theorem for upper hemicontinuous compact-valued correspondences on Polish spaces, there exists a Borel measurable selector; fix any such selector and denote it by $\mu^\star(s', w, a)$.

Define the fixed-policy evaluation operator $\mathcal{T}^{\pi, \mu}$ exactly as in the proof of Theorem 2, with $w' \sim \mu(\cdot \mid s', w, a)$. It is a monotone $\gamma$-contraction and thus has a unique fixed point $V^{\pi, \mu} = J^{\pi, \mu}$. By construction of $(\pi^\star, \mu^\star)$ we have $\mathcal{T}V^\star = \mathcal{T}^{\pi^\star, \mu^\star} V^\star$, so $V^\star$ is a fixed point of $\mathcal{T}^{\pi^\star, \mu^\star}$ and hence equals its unique fixed point: $V^\star = V^{\pi^\star, \mu^\star}$. The saddle inequalities follow from the same comparison-lemma argument as in the grid proof (monotonicity of $\mathcal{T}$ and $\mathcal{T}^{\pi, \mu}$ and contraction). Therefore $(\pi^\star, \mu^\star)$ is a stationary saddle and the min–max equalities hold. $\square$

## B.6. Proof of Theorem 3

*Proof.* We first bound the distance between an arbitrary $V$ and the fixed point $V^\star$, then bound the value of the greedy saddle pair induced by $V$.

*Residual implies value-function error.* Using the fixed-point identity $V^\star = \mathcal{T}V^\star$ and the contraction property,

$$\begin{aligned}
\|V - V^\star\|_\infty &= \|V - \mathcal{T}V^\star\|_\infty \\
&\leq \|V - \mathcal{T}V\|_\infty + \|\mathcal{T}V - \mathcal{T}V^\star\|_\infty \\
&\leq \|V - \mathcal{T}V\|_\infty + \gamma \|V - V^\star\|_\infty.
\end{aligned}$$

Rearranging yields $\|V - V^\star\|_\infty \leq \|\mathcal{T}V - V\|_\infty / (1 - \gamma)$.

*Greedy saddle pair.* Assume maximizers/minimizers in $\mathcal{T}V$ are selected to define a stationary deterministic pair $(\pi_V, \mu_V)$ such that

$$\mathcal{T}V = \mathcal{T}^{\pi_V, \mu_V} V. \tag{6}$$

(The existence of such selectors is automatic on a finite grid.) Let $V^{\pi_V, \mu_V}$ be the fixed point of the evaluation operator $\mathcal{T}^{\pi_V, \mu_V}$. Since $\mathcal{T}^{\pi_V, \mu_V}$ is a $\gamma$-contraction,

$$
\begin{aligned}
\|V^{\pi_V, \mu_V} - V\|_\infty &= \|\mathcal{T}^{\pi_V, \mu_V} V^{\pi_V, \mu_V} - V\|_\infty \\
&\leq \|\mathcal{T}^{\pi_V, \mu_V} V^{\pi_V, \mu_V} - \mathcal{T}^{\pi_V, \mu_V} V\|_\infty + \|\mathcal{T}^{\pi_V, \mu_V} V - V\|_\infty \\
&\leq \gamma \|V^{\pi_V, \mu_V} - V\|_\infty + \|\mathcal{T}V - V\|_\infty,
\end{aligned}
$$

where we used (6) in the last term. Rearranging gives $\|V^{\pi_V, \mu_V} - V\|_\infty \leq \|\mathcal{T}V - V\|_\infty / (1 - \gamma)$. Finally, by the triangle inequality,

$$
\begin{aligned}
\|V^{\pi_V, \mu_V} - V^\star\|_\infty &\leq \|V^{\pi_V, \mu_V} - V\|_\infty + \|V - V^\star\|_\infty \\
&\leq \frac{2\|\mathcal{T}V - V\|_\infty}{1 - \gamma}.
\end{aligned}
$$

$\square$

## B.7. Proof of Theorem 4

*Proof.* We provide a self-contained proof based on contraction and the greedy definition of $\pi_{\widehat{V}}$.

*Policy-evaluation operator.* For any fixed stationary agent policy $\pi$, define the associated evaluation operator

$$(\mathcal{T}^\pi V)(s, w) \triangleq \mathbb{E}_{a \sim \pi(\cdot | s, w)} \mathbb{E}_{s' \sim P(\cdot | s, a)} \Big[ \min_{w' \in \mathcal{W}} \{ \langle w', r(s, a, s') \rangle + \lambda D_\Phi(w' \mid w) + \gamma V(s', w') \} \Big]. \tag{7}$$

The proof of Theorem 1 (with the outer $\max$ removed) shows that $\mathcal{T}^\pi$ is a monotone $\gamma$-contraction under $\|\cdot\|_\infty$, hence it admits a unique fixed point $V^\pi$. By unrolling the recursion, $V^\pi(s, w)$ equals the robust value $\min_\mu J^{\pi, \mu}(s, w)$.

*Greedy policy and residual.* Recall that $\pi_{\widehat{V}}$ is greedy with respect to $\widehat{V}$ as defined in Theorem 4. By definition of the greedy maximizer,

$$\mathcal{T}\widehat{V} = \mathcal{T}^{\pi_{\hat{v}}} \widehat{V}, \tag{8}$$

so the Bellman residual satisfies

$$\varepsilon(\widehat{V}) = \|\mathcal{T}\widehat{V} - \widehat{V}\|_\infty = \|\mathcal{T}^{\pi_{\hat{v}}} \widehat{V} - \widehat{V}\|_\infty.$$

*Bounding $\|V^{\pi_{\hat{v}}} - \widehat{V}\|_\infty$.* Using the fixed-point property $V^{\pi_{\hat{v}}} = \mathcal{T}^{\pi_{\hat{v}}} V^{\pi_{\hat{v}}}$ and the contraction of $\mathcal{T}^{\pi_{\hat{v}}}$,

$$
\begin{aligned}
\|V^{\pi_{\hat{v}}} - \widehat{V}\|_\infty &= \|\mathcal{T}^{\pi_{\hat{v}}} V^{\pi_{\hat{v}}} - \widehat{V}\|_\infty \\
&\leq \|\mathcal{T}^{\pi_{\hat{v}}} V^{\pi_{\hat{v}}} - \mathcal{T}^{\pi_{\hat{v}}} \widehat{V}\|_\infty + \|\mathcal{T}^{\pi_{\hat{v}}} \widehat{V} - \widehat{V}\|_\infty \\
&\leq \gamma \|V^{\pi_{\hat{v}}} - \widehat{V}\|_\infty + \varepsilon(\widehat{V}),
\end{aligned}
$$

which rearranges to

$$\|V^{\pi_{\hat{v}}} - \widehat{V}\|_\infty \leq \frac{\varepsilon(\widehat{V})}{1 - \gamma}. \tag{9}$$

*Bounding the robust suboptimality gap.* Theorem 3 gives $\|\widehat{V} - V^\star\|_\infty \leq \varepsilon(\widehat{V})/(1 - \gamma)$. Combining this with (9) yields

$$\|V^{\pi_{\hat{v}}} - V^\star\|_\infty \leq \|V^{\pi_{\hat{v}}} - \widehat{V}\|_\infty + \|\widehat{V} - V^\star\|_\infty \leq \frac{2\varepsilon(\widehat{V})}{1 - \gamma}.$$

Finally, since $V^\star = \max_\pi \min_\mu J^{\pi, \mu}$, we have $V^{\pi_{\hat{v}}} \leq V^\star$ pointwise, so

$$0 \leq V^\star(s, w) - V^{\pi_{\hat{v}}}(s, w) \leq \frac{2\varepsilon(\widehat{V})}{1 - \gamma} \qquad \forall (s, w).$$

$\square$

**B.8. Proof of Proposition 1**

*Proof.* We prove monotonicity first, then address the limiting regimes.

*Monotonicity in $\lambda$.* Fix any bounded $V$ and any tuple $(s, w, a, s')$. For $\lambda \geq 0$ define

$$g_\lambda(w') := \langle w', r(s, a, s') \rangle + \lambda D_\Phi(w' \mid w) + \gamma V(s', w').$$

If $\lambda_2 \geq \lambda_1$, then for every $w' \in \mathcal{W}$,

$$\begin{aligned} g_{\lambda_2}(w') &= g_{\lambda_1}(w') + (\lambda_2 - \lambda_1) D_\Phi(w' \mid w) \\ &\geq g_{\lambda_1}(w'), \end{aligned}$$

since $D_\Phi(\cdot \mid \cdot) \geq 0$. Taking $\min_{w'}$, then expectation over $s'$, and finally $\max_a$ preserves the inequality, so $\mathcal{T}_{\lambda_2} V \geq \mathcal{T}_{\lambda_1} V$ pointwise. Applying value iteration from a common initialization and taking limits using contraction yields $V_{\lambda_2}^\star \geq V_{\lambda_1}^\star$.

*The regime $\lambda \to 0$.* Assume $D_\Phi$ is bounded on $\mathcal{W}$ by $D_{\max}$. Fix any bounded $V$ and any tuple $(s, w, a, s')$. Write $r := r(s, a, s')$. For every $w' \in \mathcal{W}$,

$$\begin{aligned} \langle w', r \rangle + \gamma V(s', w') &\leq \langle w', r \rangle + \lambda D_\Phi(w' \mid w) + \gamma V(s', w') \\ &\leq \langle w', r \rangle + \gamma V(s', w') + \lambda D_{\max}. \end{aligned}$$

Taking $\min_{w'}$ gives $0 \leq m_\lambda(s, w, a, s') - m_0(s, w, a, s') \leq \lambda D_{\max}$, where $m_\lambda$ denotes the inner minimum. After expectation over $s'$ and maximizing over $a$, we obtain $\|\mathcal{T}_\lambda V - \mathcal{T}_0 V\|_\infty \leq \lambda D_{\max}$. Let $V_\lambda^\star$ and $V_0^\star$ be the fixed points of $\mathcal{T}_\lambda$ and $\mathcal{T}_0$. Using $V_\lambda^\star = \mathcal{T}_\lambda V_\lambda^\star$ and $V_0^\star = \mathcal{T}_0 V_0^\star$,

$$\begin{aligned} \|V_\lambda^\star - V_0^\star\|_\infty &\leq \|\mathcal{T}_\lambda V_\lambda^\star - \mathcal{T}_\lambda V_0^\star\|_\infty + \|\mathcal{T}_\lambda V_0^\star - \mathcal{T}_0 V_0^\star\|_\infty \\ &\leq \gamma \|V_\lambda^\star - V_0^\star\|_\infty + \lambda D_{\max}. \end{aligned}$$

Rearranging yields $\|V_\lambda^\star - V_0^\star\|_\infty \leq \lambda D_{\max}/(1 - \gamma)$, which implies $V_\lambda^\star \to V_0^\star$ as $\lambda \downarrow 0$.

*The regime $\lambda \to \infty$ on a finite grid.* Assume $\mathcal{W} = \mathcal{W}_G$ is finite and $D_\Phi(w' \mid w) = 0$ if and only if $w' = w$. Define the minimum positive divergence

$$\delta_{\min} := \min_{w \in \mathcal{W}_G} \min_{w' \in \mathcal{W}_G \setminus \{w\}} D_\Phi(w' \mid w) > 0.$$

Fix any bounded $V$ and define $B_V := R_{\max} + \gamma \|V\|_\infty$. For any tuple $(s, w, a, s')$ and any $w' \neq w$, we have

$$\begin{aligned} \langle w', r(s, a, s') \rangle + \gamma V(s', w') &\geq -B_V, \\ \langle w, r(s, a, s') \rangle + \gamma V(s', w) &\leq B_V. \end{aligned}$$

Hence, for $w' \neq w$,

$$\big(\langle w', r \rangle + \gamma V(s', w')\big) + \lambda D_\Phi(w' \mid w) \geq -B_V + \lambda \delta_{\min}.$$

If $\lambda \delta_{\min} > 2B_V$, then $-B_V + \lambda \delta_{\min} > B_V$, so the unique minimizer of $w' \mapsto \langle w', r \rangle + \lambda D_\Phi(w' \mid w) + \gamma V(s', w')$ is $w' = w$. In this regime, the inner minimization "freezes" to $w_{t+1} = w_t$ and the Bellman–Isaacs operator reduces to

$$(\mathcal{T}_\infty V)(s, w) := \max_{a \in \mathcal{A}} \mathbb{E}_{s' \sim P(\cdot \mid s, a)} \big[ \langle w, r(s, a, s') \rangle + \gamma V(s', w) \big].$$

In particular, once $\lambda$ is large enough that the above condition holds uniformly over the value-iteration iterates (e.g., when $V$ is taken to be the fixed point), the game becomes equivalent to a collection of fixed-weight scalar MDPs indexed by $w$.

*A quantitative drift bound in the continuous-weight case.* For completeness, we record the standard inequality underlying the "freezing" intuition on compact $\mathcal{W}$. Fix any bounded function $f$ on $\mathcal{W}$ with $|f(w')| \leq B$. Let $w_\lambda \in \arg\min_{w' \in \mathcal{W}} f(w') + \lambda D_\Phi(w' \mid w)$. Since $w' = w$ is feasible,

$$\lambda D_\Phi(w_\lambda \mid w) \leq f(w) - f(w_\lambda) \leq 2B,$$

hence $D_\Phi(w_\lambda \mid w) \leq 2B/\lambda$. If $\Phi$ is $\sigma$-strongly convex w.r.t. $\|\cdot\|_2$, then $D_\Phi(u \mid v) \geq \frac{\sigma}{2} \|u - v\|_2^2$ and therefore $\|w_\lambda - w\|_2 \leq \sqrt{\frac{4B}{\sigma \lambda}}$. $\qquad \square$

**Theorem 6** (A global $O(1/\lambda)$ drift bound). *Assume the vector reward is bounded as $\|r(s, a, s')\|_\infty \leq R_{\max}$ for all $(s, a, s')$ and $\mathcal{W} \subseteq \Delta^d$. For any $\lambda > 0$, let $(\pi^\star, \mu^\star)$ be a saddle policy pair for the discounted game with switching penalty $\lambda$. Define the discounted drift*

$$\mathrm{DRIFT}(\pi^\star, \mu^\star) := \mathbb{E}^{\pi^\star, \mu^\star} \Big[ \sum_{t=0}^\infty \gamma^t D_\Phi(w_{t+1} \mid w_t) \Big].$$

*Then*

$$\mathrm{DRIFT}(\pi^\star, \mu^\star) \leq \frac{2R_{\max}}{(1-\gamma)\,\lambda}.$$

*Proof.* Write the game return as the sum of a weighted-return term and the drift penalty,

$$J_\lambda(\pi, \mu) = \underbrace{\mathbb{E}^{\pi, \mu} \Big[ \sum_{t=0}^\infty \gamma^t \langle w_{t+1}, r_t \rangle \Big]}_{\mathrm{WRR}(\pi, \mu)} + \lambda \underbrace{\mathbb{E}^{\pi, \mu} \Big[ \sum_{t=0}^\infty \gamma^t D_\Phi(w_{t+1} \mid w_t) \Big]}_{\mathrm{DRIFT}(\pi, \mu)}.$$

Since $w_{t+1} \in \Delta^d$ and $\|r_t\|_\infty \leq R_{\max}$, we have $|\langle w_{t+1}, r_t \rangle| \leq R_{\max}$ and thus $\mathrm{WRR}(\pi, \mu) \in [-R_{\max}/(1-\gamma),\, R_{\max}/(1-\gamma)]$ for any policies. Let $\mu^{\mathrm{noswitch}}$ denote the feasible adversary policy that keeps $w_{t+1} = w_t$ almost surely, hence $\mathrm{DRIFT}(\pi, \mu^{\mathrm{noswitch}}) = 0$ and $J_\lambda(\pi, \mu^{\mathrm{noswitch}}) = \mathrm{WRR}(\pi, \mu^{\mathrm{noswitch}}) \leq R_{\max}/(1-\gamma)$. Since the adversary minimizes, the game value satisfies

$$J_\lambda(\pi^\star, \mu^\star) = \max_\pi \min_\mu J_\lambda(\pi, \mu) \leq \max_\pi J_\lambda(\pi, \mu^{\mathrm{noswitch}}) \leq \frac{R_{\max}}{1-\gamma}.$$

On the other hand, for any $\pi$ and any $\mu$, $J_\lambda(\pi, \mu) \geq \mathrm{WRR}(\pi, \mu) \geq -R_{\max}/(1-\gamma)$ since $D_\Phi \geq 0$. In particular, $\mathrm{WRR}(\pi^\star, \mu^\star) \geq -R_{\max}/(1-\gamma)$. Combining,

$$\lambda\,\mathrm{DRIFT}(\pi^\star, \mu^\star) = J_\lambda(\pi^\star, \mu^\star) - \mathrm{WRR}(\pi^\star, \mu^\star) \leq \frac{R_{\max}}{1-\gamma} - \Big( -\frac{R_{\max}}{1-\gamma} \Big) = \frac{2R_{\max}}{1-\gamma},$$

which yields the stated bound. □

## C. Additional Technical Results

This section collects technical statements used to justify approximation choices in our solvers, including error propagation from approximate inner minimization (Appendix C.1), explicit accuracy guarantees for solving the inner minimization with a finite number of mirror/prox steps (Appendix C.2), discretization error from restricting the weight domain to a finite grid (Appendix C.3), the softmin surrogate on a finite grid (Appendix C.4), and closed-form Bregman updates (Appendix C.5), supporting the algorithmic details in Sec. 5.

### C.1. Approximate inner minimization and error propagation

Consider any approximate Bellman–Isaacs operator $\widetilde{\mathcal{T}}$ that satisfies a uniform perturbation bound

$$\|\widetilde{\mathcal{T}} V - \mathcal{T} V\|_\infty \leq \varepsilon_{\mathrm{in}} \qquad \text{for all bounded } V. \tag{10}$$

This covers, for example, computing the inner minimization up to additive error $\varepsilon_{\mathrm{in}}$ (uniformly over $(s, w, a, s')$).

**Lemma 3.** *Let $\mathcal{T}$ be a $\gamma$-contraction and $\widetilde{\mathcal{T}}$ satisfy (10). Let $V^\star$ and $\widetilde{V}^\star$ be the unique fixed points of $\mathcal{T}$ and $\widetilde{\mathcal{T}}$, respectively. Then*

$$\|\widetilde{V}^\star - V^\star\|_\infty \leq \frac{\varepsilon_{\mathrm{in}}}{1-\gamma}.$$

*Proof.* Using $\widetilde{V}^\star = \widetilde{\mathcal{T}} \widetilde{V}^\star$ and $V^\star = \mathcal{T} V^\star$,

$$\begin{aligned}
\|\widetilde{V}^\star - V^\star\|_\infty &= \|\widetilde{\mathcal{T}} \widetilde{V}^\star - \mathcal{T} V^\star\|_\infty \\
&\leq \|\widetilde{\mathcal{T}} \widetilde{V}^\star - \mathcal{T} \widetilde{V}^\star\|_\infty + \|\mathcal{T} \widetilde{V}^\star - \mathcal{T} V^\star\|_\infty \\
&\leq \varepsilon_{\mathrm{in}} + \gamma \|\widetilde{V}^\star - V^\star\|_\infty.
\end{aligned}$$

Rearranging proves the claim. □

## C.2. Inner minimization with $J$ mirror/prox steps

In practice we often approximate the inner minimum by running a small number of mirror/prox updates. This subsection records a simple strong-convexity regime under which the inner problem is well-conditioned and the resulting approximation error decays geometrically in the number of inner steps.

Fix any tuple $(s, w, a, s')$ and a value function $V$. Define the inner objective over $u \in \mathcal{W}$:

$$F(u) := \langle u, r(s, a, s') \rangle + \lambda D_\Phi(u \mid w) + \gamma V(s', u).$$

Assume that $u \mapsto V(s', u)$ is differentiable and $L_V$-smooth on $\mathcal{W}$ w.r.t. $\| \cdot \|_2$, and that $\Phi$ is $\sigma$-strongly convex and $L_\Phi$-smooth on $\mathcal{W}$ w.r.t. $\| \cdot \|_2$. Then $u \mapsto \gamma V(s', u)$ is $\gamma L_V$-smooth and $\gamma L_V$-weakly convex, so $F$ is $\mu$-strongly convex with

$$\mu := \lambda \sigma - \gamma L_V \qquad (\text{requiring } \lambda \sigma > \gamma L_V),$$

and $L$-smooth with $L := \lambda L_\Phi + \gamma L_V$.

**Lemma 4.** *Under the above assumptions and $\mu > 0$, let $u^\star \in \arg\min_{u \in \mathcal{W}} F(u)$ and let $(u_j)_{j \geq 0}$ be the iterates of projected gradient descent*

$$u_{j+1} = \Pi_\mathcal{W}\big(u_j - \tfrac{1}{L} \nabla F(u_j)\big).$$

*Then for all $J \geq 0$,*

$$F(u_J) - F(u^\star) \leq \Big(1 - \tfrac{\mu}{L}\Big)^J \cdot \tfrac{L}{2} \|u_0 - u^\star\|_2^2.$$

*Proof.* This is the standard gradient-descent rate for $\mu$-strongly convex and $L$-smooth functions. $\qquad\square$

A uniform inner accuracy bound $\varepsilon_{\text{in}}(J)$ can be obtained by upper bounding $\|u_0 - u^\star\|_2$ by the diameter of $\mathcal{W}$. Combining with Lemma 3 yields an explicit tradeoff between the number of inner steps and the resulting fixed-point error of the overall Bellman iteration.

## C.3. Discretization error from restricting $\mathcal{W}$ to a grid

In tabular solvers and evaluation routines we implement the weight domain by a finite grid $\mathcal{W}_G \subseteq \mathcal{W}$. This defines an approximate Bellman–Isaacs operator

$$(\mathcal{T}_G V)(s, w) := \max_{a \in \mathcal{A}} \mathbb{E}_{s' \sim P(\cdot|s,a)}\Big[ \min_{u \in \mathcal{W}_G} \{\langle u, r(s, a, s') \rangle + \lambda D_\Phi(u \mid w) + \gamma V(s', u)\}\Big].$$

Since the minimization is over a subset, we always have $\mathcal{T}_G V \geq \mathcal{T} V$ pointwise. The next proposition gives a simple Lipschitz-based discretization bound.

**Proposition 2.** *Let $\delta := \sup_{u \in \mathcal{W}} \min_{v \in \mathcal{W}_G} \|u - v\|_2$ be the covering radius of the grid. Assume that $\|r(s, a, s')\|_\infty \leq R_{\max}$ for all $(s, a, s')$, that for each $s$ the map $u \mapsto V(s, u)$ is $L_V$-Lipschitz on $\mathcal{W}$ w.r.t. $\| \cdot \|_2$, and that for each $w$ the map $u \mapsto D_\Phi(u \mid w)$ is $L_D$-Lipschitz on $\mathcal{W}$. Then for any bounded $V$,*

$$0 \leq (\mathcal{T}_G V)(s, w) - (\mathcal{T} V)(s, w) \leq \delta\big(\sqrt{d}\, R_{\max} + \lambda L_D + \gamma L_V\big) \qquad \forall (s, w).$$

*Consequently, letting $V^\star$ and $V_G^\star$ be the fixed points of $\mathcal{T}$ and $\mathcal{T}_G$,*

$$\|V_G^\star - V^\star\|_\infty \leq \frac{\delta\big(\sqrt{d}\, R_{\max} + \lambda L_D + \gamma L_V\big)}{1 - \gamma}.$$

*Proof.* Fix $(s, w, a, s')$ and let $u^\star \in \arg\min_{u \in \mathcal{W}}$ of the inner objective. Pick any grid point $v \in \mathcal{W}_G$ with $\|u^\star - v\|_2 \leq \delta$. By $\|r\|_\infty \leq R_{\max}$ we have $|\langle u^\star, r \rangle - \langle v, r \rangle| \leq \|u^\star - v\|_1 \|r\|_\infty \leq \sqrt{d}\, R_{\max} \delta$. The Lipschitz assumptions further give $|D_\Phi(u^\star \mid w) - D_\Phi(v \mid w)| \leq L_D \delta$ and $|V(s', u^\star) - V(s', v)| \leq L_V \delta$. Therefore the inner objective at $v$ is at most $\delta(\sqrt{d}\, R_{\max} + \lambda L_D + \gamma L_V)$ larger than at $u^\star$. Taking $\min_{u \in \mathcal{W}_G}$, then expectation in $s'$, and finally $\max_a$ yields the pointwise bound. The fixed-point bound follows from Lemma 3 with $\widetilde{\mathcal{T}} = \mathcal{T}_G$. $\qquad\square$

## C.4. Softmin smoothing on a finite weight grid

In deep implementations it is sometimes convenient to replace $\min_{w' \in \mathcal{W}_G}$ by a differentiable surrogate. Let $\mathcal{W}_G = \{w^{(1)}, \ldots, w^{(M)}\}$ be a finite grid and let $x_i \in \mathbb{R}$ denote the inner objective evaluated at $w^{(i)}$. Define the soft minimum for $\tau > 0$:

$$\mathrm{softmin}_\tau(x_1, \ldots, x_M) := -\tau \log\Big( \sum_{i=1}^M e^{-x_i/\tau} \Big).$$

**Lemma 5.** *For any $x \in \mathbb{R}^M$ and $\tau > 0$,*

$$\min_i x_i - \tau \log M \le \mathrm{softmin}_\tau(x) \le \min_i x_i.$$

*Proof.* Let $m := \min_i x_i$. Then $\sum_{i=1}^M e^{-x_i/\tau} \ge e^{-m/\tau}$, hence $\mathrm{softmin}_\tau(x) = -\tau \log \sum_i e^{-x_i/\tau} \le -\tau \log e^{-m/\tau} = m$. Also, since $x_i \ge m$ for all $i$, we have $e^{-x_i/\tau} \le e^{-m/\tau}$ and therefore $\sum_i e^{-x_i/\tau} \le M e^{-m/\tau}$. This implies $\mathrm{softmin}_\tau(x) \ge -\tau \log(Me^{-m/\tau}) = m - \tau \log M$. $\square$

Lemma 5 implies that replacing the exact $\min$ by $\mathrm{softmin}_\tau$ perturbs each Bellman backup by at most $\tau \log |\mathcal{W}_G|$ (in absolute value), so Lemma 3 yields an induced value-function error of at most $(\tau \log |\mathcal{W}_G|)/(1-\gamma)$.

## C.5. Closed-form Bregman updates for the inner minimization

When the continuation value is convex (e.g., a max-linear "convex critic" in $w$), the inner minimization

$$\min_{w' \in \mathcal{W}} \langle w', g \rangle + \lambda D_\Phi(w' \mid w)$$

admits efficient mirror/prox updates. We record two special cases that are used in practice.

*KL-Bregman (exponentiated gradient).* Let $\mathcal{W} = \Delta_\varepsilon^d := \{w \in \Delta^d : w_i \ge \varepsilon\}$ and $D_\Phi = D_{\mathrm{KL}}$. Ignoring the lower-bound constraint temporarily, the KKT conditions for minimizing $\langle w', g \rangle + \lambda \sum_i w'_i \log \frac{w'_i}{w_i}$ over $\Delta^d$ give $\log \frac{w'_i}{w_i} = -(g_i + c)/\lambda$ for a normalizing constant $c$, i.e.,

$$w'_i \propto w_i \exp\big( -g_i/\lambda \big). \tag{11}$$

Projecting back to $\Delta_\varepsilon^d$ can be implemented by clipping $w'$ to $[\varepsilon, 1]$ followed by renormalization.

*Squared Euclidean divergence (projected gradient).* If $D_\Phi(w' \mid w) = \frac{1}{2}\|w' - w\|_2^2$ on a convex compact $\mathcal{W}$, then $\arg\min_{w' \in \mathcal{W}} \langle w', g \rangle + \frac{\lambda}{2}\|w' - w\|_2^2$ equals the Euclidean projection

$$w' = \Pi_{\mathcal{W}}\big( w - \tfrac{1}{\lambda} g \big). \tag{12}$$

## C.6. Regularity induced by the Bregman switching cost

The switching penalty $\lambda D_\Phi(w' \mid w)$ can be viewed as a proximal regularizer on the inner minimization. In addition to enforcing time consistency, it gives quantitative stability properties that are useful for approximation and learning.

Fix any convex function $g : \mathcal{W} \to \mathbb{R}$ and define the Bregman prox mapping

$$\mathrm{Prox}_{g,\lambda}(w) \in \arg\min_{u \in \mathcal{W}} \; g(u) + \lambda D_\Phi(u \mid w). \tag{13}$$

When $g(u) = \langle u, r(s, a, s') \rangle + \gamma V(s', u)$ and $V(s', \cdot)$ is convex, the inner minimization in our Bellman–Isaacs operator is exactly of the form (13).

**Proposition 3** (Uniqueness of the inner minimizer). *Assume $\Phi$ is $\mu_\Phi$-strongly convex on $\mathcal{W}$ w.r.t. $\|\cdot\|_2$, i.e., $D_\Phi(u \mid v) \ge \frac{\mu_\Phi}{2}\|u - v\|_2^2$. If $g$ is convex, then the objective in (13) is $(\lambda\mu_\Phi)$-strongly convex in $u$ and therefore admits a unique minimizer.*

*Proof.* The map $u \mapsto D_\Phi(u \mid w)$ is $\mu_\Phi$-strongly convex when $\Phi$ is $\mu_\Phi$-strongly convex. Adding a convex term preserves strong convexity. $\square$

**Proposition 4** (Lipschitz dependence on the current weight). *Assume $g$ is convex and $\Phi$ is $\mu_\Phi$-strongly convex and $L_\Phi$-smooth on $\mathcal{W}$ w.r.t. $\|\cdot\|_2$ (so $\|\nabla\Phi(u) - \nabla\Phi(v)\|_2 \leq L_\Phi\|u - v\|_2$). Let $u = \mathrm{Prox}_{g,\lambda}(w)$ and $\tilde{u} = \mathrm{Prox}_{g,\lambda}(\tilde{w})$. Then*

$$\|u - \tilde{u}\|_2 \leq \frac{L_\Phi}{\mu_\Phi}\|w - \tilde{w}\|_2.$$

*Proof.* Let $s \in \partial g(u)$ and $\tilde{s} \in \partial g(\tilde{u})$ be subgradients satisfying the first-order optimality conditions $s + \lambda(\nabla\Phi(u) - \nabla\Phi(w)) = 0$ and $\tilde{s} + \lambda(\nabla\Phi(\tilde{u}) - \nabla\Phi(\tilde{w})) = 0$. Subtracting and taking the inner product with $u - \tilde{u}$ gives

$$\langle s - \tilde{s},\, u - \tilde{u}\rangle + \lambda\langle\nabla\Phi(u) - \nabla\Phi(\tilde{u}),\, u - \tilde{u}\rangle = \lambda\langle\nabla\Phi(w) - \nabla\Phi(\tilde{w}),\, u - \tilde{u}\rangle.$$

By monotonicity of subgradients of a convex function, $\langle s - \tilde{s}, u - \tilde{u}\rangle \geq 0$. Strong convexity of $\Phi$ implies $\langle\nabla\Phi(u) - \nabla\Phi(\tilde{u}), u - \tilde{u}\rangle \geq \mu_\Phi\|u - \tilde{u}\|_2^2$. Finally, by Cauchy–Schwarz and smoothness of $\Phi$, the right-hand side is at most $\lambda L_\Phi\|w - \tilde{w}\|_2\|u - \tilde{u}\|_2$. Canceling $\lambda$ and $\|u - \tilde{u}\|_2$ yields the claim. $\qquad\square$

**Proposition 5** (Sensitivity to perturbations of the inner objective). *Let $g$ and $\tilde{g}$ be convex functions on $\mathcal{W}$ satisfying $\sup_{u\in\mathcal{W}}|g(u) - \tilde{g}(u)| \leq \varepsilon$. Let $u = \mathrm{Prox}_{g,\lambda}(w)$ and $\tilde{u} = \mathrm{Prox}_{\tilde{g},\lambda}(w)$ for the same $w$. If $\Phi$ is $\mu_\Phi$-strongly convex on $\mathcal{W}$, then*

$$\|u - \tilde{u}\|_2 \leq \sqrt{\frac{4\varepsilon}{\lambda\mu_\Phi}}.$$

*Proof.* Since $u$ minimizes $g(u) + \lambda D_\Phi(u\mid w)$, $g(u) + \lambda D_\Phi(u\mid w) \leq g(\tilde{u}) + \lambda D_\Phi(\tilde{u}\mid w)$. Using $g \leq \tilde{g} + \varepsilon$ and $\tilde{g} \leq g + \varepsilon$ yields $\tilde{g}(u) + \lambda D_\Phi(u\mid w) \leq \tilde{g}(\tilde{u}) + \lambda D_\Phi(\tilde{u}\mid w) + 2\varepsilon$, so $u$ is a $2\varepsilon$-suboptimal point for the strongly convex objective $u \mapsto \tilde{g}(u) + \lambda D_\Phi(u\mid w)$. Strong convexity implies $2\varepsilon \geq \frac{\lambda\mu_\Phi}{2}\|u - \tilde{u}\|_2^2$, which gives the stated bound. $\qquad\square$

*Remark.* When $D_\Phi$ is jointly convex in $(u, w)$ (e.g., squared Euclidean distance or KL on $\Delta_\varepsilon^d$), the mapping $w \mapsto \min_{u\in\mathcal{W}}\{g(u) + \lambda D_\Phi(u\mid w)\}$ preserves convexity in $w$ whenever $g$ is convex. Therefore, if each continuation slice $V(s', \cdot)$ is convex, the Bellman–Isaacs operator preserves convexity in $w$.

# D. Further theoretical analyses

### D.1. Inner vs. outer robustness: a concrete separation

The main paper emphasizes that putting the preference adversary *inside* the Bellman backup yields a strictly stronger (and time-consistent) robustness notion than episode-level (outer) max–min. We record a simple two-step construction showing that an outer-optimal policy can be arbitrarily suboptimal under the step-wise (inner) adversary when $\lambda = 0$.

**Proposition 6** (A two-step inner–outer gap). *Fix any discount $\gamma \in (0, 1)$ and any $\varepsilon \in (0, 1)$. Consider a two-objective ($d = 2$) MDP with initial state $s_0$, intermediate states $s_1, s_2$, and an absorbing terminal state $s_\mathrm{T}$. At $s_0$, action $a_\mathrm{safe}$ transitions deterministically to $s_\mathrm{T}$ with reward vector $r = (1 - \varepsilon, 1 - \varepsilon)$, while action $a_\mathrm{risky}$ transitions to $s_1$ or $s_2$ with probability $1/2$ each with reward vector $r = (0, 0)$. At $s_1$ (resp. $s_2$) there is a single action transitioning to $s_\mathrm{T}$ with reward vector $r = (2/\gamma, 0)$ (resp. $r = (0, 2/\gamma)$). Assume $\mathcal{W} = \Delta^2$ and $\lambda = 0$.*

*Let $V_\mathrm{outer}^\star$ denote the episode-level objective where the adversary chooses a single weight $w \in \Delta^2$ at time 0 and the same $w$ is used to scalarize all rewards. Let $V_\mathrm{inner}^\star$ denote our step-wise inner objective where the adversary may choose $w_{t+1}$ after each transition and uses it to scalarize $r_t$. Then the unique outer-optimal policy takes $a_\mathrm{risky}$ at $s_0$ and achieves value $V_\mathrm{outer}^\star(s_0, w_0) = 1$ for every $w_0 \in \Delta^2$; whereas under the inner objective the robust value of $a_\mathrm{risky}$ is 0 and the inner-optimal policy takes $a_\mathrm{safe}$ with value $V_\mathrm{inner}^\star(s_0, w_0) = 1 - \varepsilon$. In particular, the outer-optimal policy suffers an inner-robust suboptimality gap of $1 - \varepsilon$.*

*Proof.* For the outer objective, the scalarized discounted return of $a_\mathrm{safe}$ equals $\langle w, (1 - \varepsilon, 1 - \varepsilon)\rangle = 1 - \varepsilon$ for every $w \in \Delta^2$. For $a_\mathrm{risky}$, the only nonzero reward appears at time $t = 1$ and equals either $(2/\gamma, 0)$ or $(0, 2/\gamma)$. Under a fixed weight $w = (w_1, w_2)$, the expected scalar reward at $t = 1$ is $\frac{1}{2}\cdot\frac{2}{\gamma}w_1 + \frac{1}{2}\cdot\frac{2}{\gamma}w_2 = \frac{1}{\gamma}$. Discounting by $\gamma$ yields outer value 1, uniformly over $w$. Hence the outer adversary cannot reduce the value below 1 and the agent strictly prefers $a_\mathrm{risky}$.

For the inner objective with $\lambda = 0$, the adversary chooses $w_2$ after observing whether the reward at $t = 1$ is $(2/\gamma, 0)$ or $(0, 2/\gamma)$. In the first case it can choose $w_2 = (0, 1)$, and in the second case $w_2 = (1, 0)$, making the scalar reward at

$t = 1$ equal to $0$ in either branch. Thus the robust value of $a_{\text{risky}}$ is $0$, while the robust value of $a_{\text{safe}}$ remains $1 - \varepsilon$ since $\langle w_1, (1 - \varepsilon, 1 - \varepsilon) \rangle = 1 - \varepsilon$ for all $w_1 \in \Delta^2$. $\qquad \square$

### D.2. A tabular sample-complexity remark for the grid implementation

In the tabular/grid regime used for our exact value iteration and BR-$K$ evaluation, the interaction can be viewed as a discounted zero-sum stochastic game on the augmented state space $\mathcal{X} = \mathcal{S} \times \mathcal{W}_G$, where $\mathcal{W}_G$ is a finite grid of size $M$. The agent has $|\mathcal{A}|$ actions, and the adversary chooses the next weight from $\mathcal{W}_G$.

Under a generative model for this finite game, standard model-based algorithms for discounted zero-sum stochastic games yield sample complexity polynomial in $|\mathcal{S}|, |\mathcal{A}|, M, (1 - \gamma)^{-1}$ and $\varepsilon^{-1}$ for computing an $\varepsilon$-optimal value function (and thus an $\varepsilon$-optimal policy). Concretely, treating the adversary's grid choice as an action yields an effective action count on the order of $|\mathcal{A}| \cdot M$ and a state count $|\mathcal{S}| \cdot M$, so the leading dependence is of the form $\widetilde{O}(|\mathcal{S}| \, |\mathcal{A}| \, M^2 \, (1 - \gamma)^{-3} \varepsilon^{-2})$ up to logarithmic factors. This perspective can be used to import tabular learning guarantees from the discounted Markov-game literature.

## E. Experimental Setup and Implementation Details

This section summarizes environments, weight domains, metrics, and the BR/BR-$K$ evaluation protocol used in our experiments (supporting Sec. 6–7).

### E.1. Environments and vector rewards

The main experiments use two-objective environments ($d = 2$); Appendix F.3 adds 4-objective and 6-objective stress tests. The agent observes the environment state $s_t$. The adversary (when trained as a best response) observes a tuple containing $s_{t+1}$ and the current weight $w_t$; in some ablations we additionally provide the realized vector reward $r_t$.

*MountainCar variants.* Timespeed and Timemove are multi-objective variants of the classic discrete-action MountainCar domain. The environment has a 2D continuous state (position, velocity) and 3 discrete actions. Rewards are 2D vectors constructed from the standard time penalty ($-1$ per step) and a second objective: Timemove uses a movement-related penalty objective (merged forward/reverse penalties), while Timespeed uses a speed-related objective. Episodes are truncated at 200 environment steps.

*DeepSeaTreasure.* DeepSeaTreasure is a 2D grid world with 4 actions (up/down/left/right) and a 2D discrete state (the agent's grid coordinates). The reward is two-dimensional: a time penalty of $-1$ per step and a nonnegative treasure value collected upon reaching a treasure. Episodes terminate upon reaching a treasure; we also cap rollouts at 200 steps for evaluation.

### E.2. Weight domain, divergences, and drift statistics

Unless stated otherwise, the weight domain is a uniform grid of 21 points on the 2D simplex. When using KL divergence, we restrict to an interior simplex $\Delta_\varepsilon^2$ to avoid boundary singularities; the deep experiments use $\varepsilon = 10^{-3}$ and the toy counterexample uses $\varepsilon = 10^{-4}$. We initialize with $w_0 = (0.5, 0.5)$ except in the initial-weight sensitivity study.

For any rollout, we compute the discounted drift

$$\text{DRIFT} := \sum_{t \geq 0} \gamma^t D_\Phi(w_{t+1} \mid w_t),$$

the weighted reward return

$$\text{WRR} := \sum_{t \geq 0} \gamma^t \langle w_{t+1}, r_t \rangle,$$

and the (discounted) vector return $R := \sum_{t \geq 0} \gamma^t r_t$. From $R \in \mathbb{R}^2$ we report

$$\min_{\text{obj}} := \min\{R_1, R_2\},$$

$$\text{GAP} := \max\{R_1, R_2\} - \min\{R_1, R_2\}.$$

The game return optimized in the zero-sum game is

$$\text{GAME} := \sum_{t \geq 0} \gamma^t \Big( \langle w_{t+1}, r_t \rangle + \lambda \, D_\Phi(w_{t+1} \mid w_t) \Big).$$

We report $(\text{WRR}, \text{DRIFT})$ to make the performance–drift tradeoff explicit; for any fixed $\lambda$, $\text{GAME} = \text{WRR} + \lambda \cdot \text{DRIFT}$.

### E.3. Networks and training loops

Deep experiments use value-based learners for both the agent and (when applicable) the BR adversary. Because all environments have low-dimensional observations, we use fully connected neural networks throughout.

*Agent.* The agent is a DQN-style learner whose Q-network takes as input the environment state and the current weight, and outputs Q-values over environment actions. We implement the target-network and replay-buffer mechanism so that the update corresponds to a fitted Bellman backup of the form described in Section 4 of the main paper. When the algorithm requires an inner minimization over $w'$, we either (i) enumerate a discrete weight grid or (ii) apply the softmin surrogate in Appendix C.4.

*BR adversary.* In BR and BR-K evaluation, the adversary is trained as a DQN-style learner whose action space is the same discrete weight grid used for evaluation. The adversary observes $(s_{t+1}, w_t)$ and (unless removed by ablation) the realized reward $r_t$. The adversary receives the negative stage payoff $-(\langle w_{t+1}, r_t \rangle + \lambda \, D_\Phi(w_{t+1} \mid w_t))$, so maximizing its own return corresponds to minimizing GAME. The adversary is trained against a frozen agent checkpoint.

*Architectural parity across methods.* For all deep comparisons, we control for representational capacity by matching the backbone network class (MLP with the same depth and width) across our method and preference-conditioned baselines. Exact layer widths, learning-rate schedules, replay-buffer sizes, and exploration schedules are specified in the released configuration files.

### E.4. BR and BR-K adversary training

The strong-adversary evaluations in the main paper use best-response adversaries trained against a fixed agent policy. We summarize the protocol implemented in our codebase.

*Best-response objective and information structure.* During BR training, the agent policy is frozen. The adversary is trained to *minimize* GAME induced by the agent—equivalently, to maximize the negative of the game return. The default adversary observation is $(s_{t+1}, w_t, r_t)$; the "state+weight" ablation removes $r_t$.

*BR-K (multi-restart min-over-$K$).* A single BR training run can produce an under-trained adversary that overestimates robustness. BR-K reduces this variance by training $K$ independent BR adversaries (different random seeds / initializations) against the same fixed agent checkpoint. Among these $K$ adversaries, we select the one that achieves the *lowest* mean WRR against the agent over evaluation rollouts, and we report that adversary's $(\text{WRR}, \text{DRIFT})$ together with its associated GAP/min_obj statistics.

*Budgets used in figures.* The default strong-adversary protocol uses BR training budget `br_steps` $= 200{,}000$ and $K = 3$ restarts. The BR-strength ablation varies `br_steps` $\in \{25{,}000, 50{,}000, 100{,}000, 200{,}000\}$ while keeping $K = 3$. All BR evaluations use 200 episodes with a rollout cap of 200 steps.

## F. Additional Experimental Results

This section reports secondary plots and diagnostics referenced in the main paper, including initial-weight sensitivity, numerical tables computed from evaluation logs, additional stress tests, and ablations that support Sec. 7.

### F.1. Initial-weight sensitivity

The step-wise opponent is initialized at an initial preference $w_0$, which can materially affect how quickly it finds a harmful reweighting trajectory. To make this sensitivity explicit, Fig. 3 sweeps $w_0$ over a coarse grid (here, the first-objective weight $w_0[1] \in \{0.1, 0.3, 0.5, 0.7, 0.9\}$). For each $w_0$, we evaluate the learned agent against BR-$K$ and record the resulting BR-$K$ metrics.

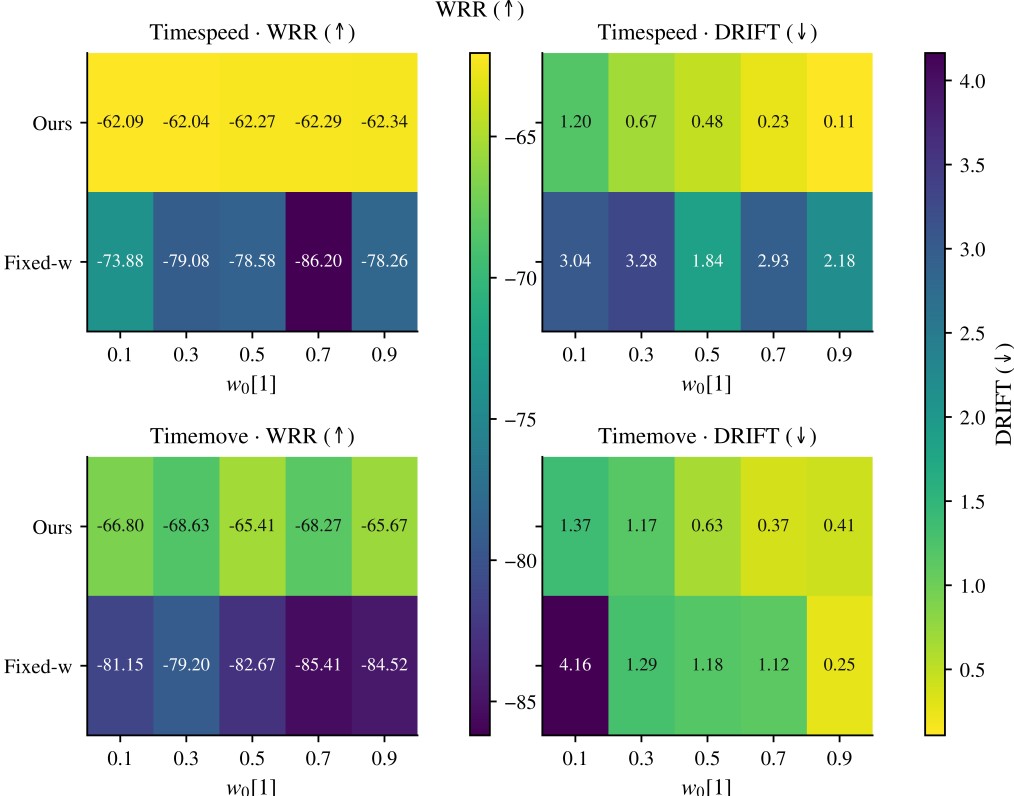

*Figure 3.* Initial-weight sensitivity under BR-$K$ ($K = 3$, `br_steps = 100`k) at $\lambda = 1.0$. Rows correspond to methods (ours vs. a fixed-weight baseline), and columns sweep the initial adversary weight $w_0$ (shown as $w_0[1]$ on the $x$-axis). The left panels report **WRR** (higher is better) and the right panels report **DRIFT** (lower is better), separately for Timespeed (top) and Timemove (bottom). Across this sweep, our method maintains consistently higher WRR with substantially lower DRIFT, while the fixed-weight baseline is both less robust and more sensitive to the adversary's initialization.

## F.2. Numerical summary for the MountainCar panels of Figure 2

Table 4 provides the numeric values underlying Fig. 2, including the Timespeed and Timemove rows referenced in the main text. For convenience, we highlight two takeaways from the MountainCar entries: (i) our method achieves the best (least negative) WRR among the compared baselines, and (ii) this improvement does *not* require large preference drift—our DRIFT is substantially smaller than preference-conditioned baselines (and comparable to or smaller than the fixed-weight baseline).

*Table 4.* Numerical counterpart of Fig. 2 (step-wise best-response evaluation at $\lambda = 1.0$ and $w_0 = (0.5, 0.5)$). Each entry is mean $\pm$ standard error across independent agent seeds. We estimate the *worst-case* by training $K = 3$ independent step-wise best-response adversaries against the same frozen agent and taking the lowest achieved WRR (reporting drift/gap for that same adversary).

| Env | Method | WRR ($\uparrow$) | DRIFT ($\downarrow$) | GAP ($\downarrow$) |
|---|---|---|---|---|
| Timespeed | Ours | -63.13 $\pm$ 0.56 | 0.428 $\pm$ 0.014 | 81.3 $\pm$ 0.9 |
| Timespeed | CN-DQN | -73.33 $\pm$ 2.31 | 2.246 $\pm$ 0.501 | 109.6 $\pm$ 4.1 |
| Timespeed | Envelope | -77.79 $\pm$ 2.37 | 1.965 $\pm$ 0.618 | 104.5 $\pm$ 5.1 |
| Timespeed | PCN | -80.92 $\pm$ 0.43 | 3.980 $\pm$ 0.382 | 98.4 $\pm$ 0.0 |
| Timespeed | Fixed-w | -74.89 $\pm$ 3.38 | 2.895 $\pm$ 0.728 | 97.5 $\pm$ 2.1 |
| Timemove | Ours | -68.56 $\pm$ 2.80 | 1.051 $\pm$ 0.235 | 9.8 $\pm$ 2.9 |
| Timemove | CN-DQN | -76.30 $\pm$ 0.82 | 4.875 $\pm$ 0.774 | 32.2 $\pm$ 4.4 |
| Timemove | Envelope | -79.87 $\pm$ 1.35 | 0.536 $\pm$ 0.069 | 82.5 $\pm$ 3.7 |
| Timemove | PCN | -84.51 $\pm$ 0.00 | 0.875 $\pm$ 0.000 | 86.6 $\pm$ 0.0 |
| Timemove | Fixed-w | -82.86 $\pm$ 1.74 | 0.827 $\pm$ 0.239 | 86.5 $\pm$ 0.0 |
| DeepSeaTreasure | Ours | -0.74 $\pm$ 0.00 | 0.270 $\pm$ 0.000 | 1.7 $\pm$ 0.0 |
| DeepSeaTreasure | CN-DQN | -34.47 $\pm$ 3.71 | 102.703 $\pm$ 32.762 | 63.0 $\pm$ 0.0 |
| DeepSeaTreasure | Envelope | -57.42 $\pm$ 2.67 | 3.188 $\pm$ 1.561 | 63.0 $\pm$ 0.0 |
| DeepSeaTreasure | PCN | -23.81 $\pm$ 18.66 | 0.385 $\pm$ 0.046 | 32.1 $\pm$ 16.0 |
| DeepSeaTreasure | Fixed-w | -8.04 $\pm$ 2.07 | 0.443 $\pm$ 0.036 | 8.3 $\pm$ 2.0 |

## F.3. Additional higher-dimensional and continuous-adversary stress tests

We include three additional stress tests that address the empirical scope of the method without changing the main experimental protocol. First, Table 5 evaluates higher-dimensional objective spaces. We use the same definitions of WRR and DRIFT as in the main paper. Since the main text defines GAP for two objectives, the $d > 2$ rows report

$$\text{GAP} := \max_i R_i - \min_i R_i,$$

where $R$ is the discounted vector return. These stress tests are supporting evidence rather than a replacement for the main benchmark suite.

*Table 5.* Higher-dimensional stress tests. WRR is higher-better; DRIFT and GAP are lower-better.

| Benchmark | Method | WRR ($\uparrow$) | DRIFT ($\downarrow$) | GAP ($\downarrow$) |
|---|---|---|---|---|
| 4D MO-Lunar-Lander | Ours | **-3.0473** | **3.1751** | **31.9227** |
| 4D MO-Lunar-Lander | CN-DQN | -6.7256 | 108.4052 | 40.5880 |
| 4D MO-Lunar-Lander | Envelope | -5.1989 | 34.6925 | 33.1273 |
| 4D MO-Lunar-Lander | PCN | -53.5297 | 13.1815 | 88.7153 |
| 4D MO-Lunar-Lander | Fixed-$w$ | -16.6088 | 126.8366 | 85.9096 |
| 6D Fruit-Tree | Ours | **6.6308** | **3.4498** | **2.5217** |
| 6D Fruit-Tree | CN-DQN | 4.6773 | 17.2689 | 5.8035 |
| 6D Fruit-Tree | Envelope | 6.5321 | 3.7428 | 7.7547 |
| 6D Fruit-Tree | PCN | 3.6501 | 10.2136 | 4.7831 |
| 6D Fruit-Tree | Fixed-$w$ | 3.9963 | 10.8526 | 2.7528 |

Second, we check whether the discrete-grid BR-$K$ conclusions are overturned by a continuous adversary. Table 6 keeps the DeepSeaTreasure environment, post-transition observation, and stage objective fixed, but replaces the finite-grid adversary action with a continuous TD3 adversary that outputs $w_{t+1}$ on the simplex. Under this check, our method remains stronger than the fixed-weight baseline.

*Table 6.* Continuous TD3 best-response adversary on DeepSeaTreasure.

| Method | WRR ($\uparrow$) | DRIFT ($\downarrow$) | GAP ($\downarrow$) |
|---|---|---|---|
| Ours | **-1.0000** | **0.5821** | **1.7000** |
| Fixed-$w$ | -5.8520 | 0.6931 | 5.8520 |

Third, Table 7 varies the BR-$K$ restart count and the best-response training budget on a fixed Timespeed/Ours checkpoint. This is a fixed-checkpoint sensitivity check rather than a full cross-method re-ranking. The worst-of-$K$ estimate changes only slightly from $K = 3$ to $K = 10$, and increasing the BR budget from 200k to 500k does not qualitatively alter the conclusion for this checkpoint.

*Table 7.* BR-$K$ sensitivity on a fixed Timespeed/Ours checkpoint.

| $K$ | 3 | 5 | 10 |
|---|---|---|---|
| Worst-of-$K$ mean WRR ($\uparrow$) | -62.7043 | -62.7393 | -62.7412 |

| BR budget | WRR ($\uparrow$) | DRIFT ($\downarrow$) | GAP ($\downarrow$) |
|---|---|---|---|
| 200k | -62.7393 | 0.5596 | 81.3547 |
| 500k | -62.0274 | 0.4649 | 79.7071 |

### F.4. Ablations: opponent observation and divergence geometry

We include two diagnostic checks that help interpret robustness results and clarify which modeling/evaluation choices matter. Figure 4 summarizes both ablations in a compact "dashboard".

**(a) Divergence geometry (KL vs. $\ell_2$).** We compare entropic (KL) and squared-$\ell_2$ switching costs in the inner minimization. The markers show the resulting BR-$K$ metrics for Timespeed and Timemove (ours only), and the annotated $\Delta$ reports $\ell_2$ *minus KL*. Negative $\Delta$ in the return plot means $\ell_2$ yields lower WRR than KL (worse), while positive $\Delta$ in the drift plot means $\ell_2$ induces larger DRIFT.

**(b) Opponent observation (strong vs. weak).** We compare the "strong" observation model obs $= (s_{t+1}, w_t, r_t)$ (used for robustness claims) to a weaker model obs $= (s_{t+1}, w_t)$ that omits the realized reward. The annotated $\Delta$ reports *weak minus strong*. As expected, restricting the opponent observation generally makes the adversary less effective (higher WRR), which is why we adopt the stronger information structure for our main evaluation.

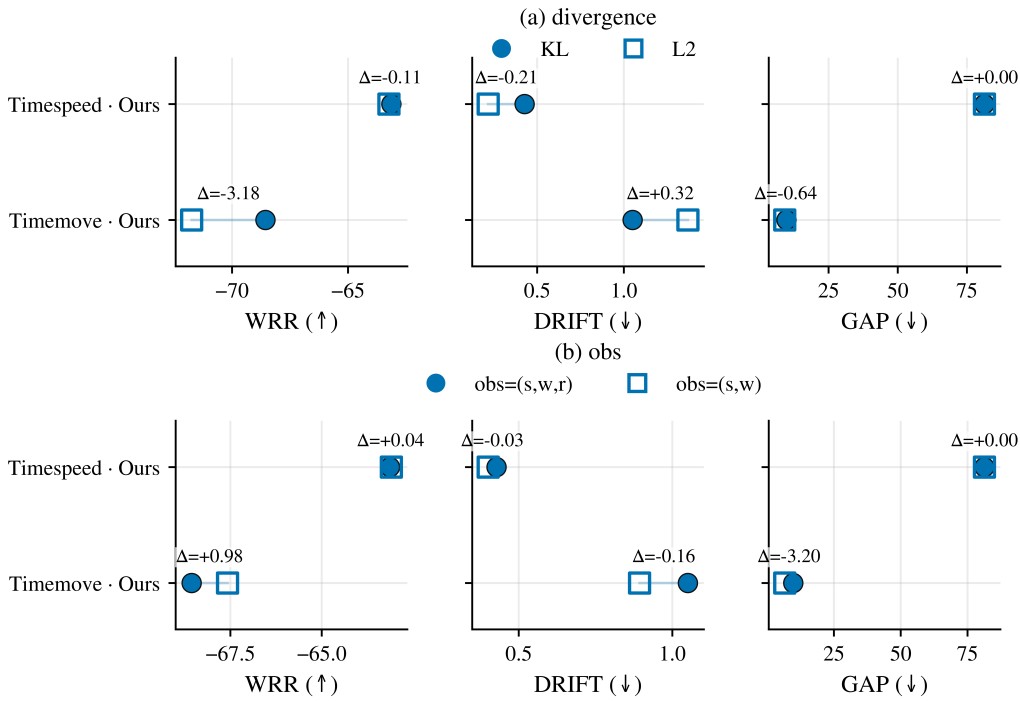

*Figure 4.* Ablation dashboard under BR-$K$ at $\lambda = 1.0$ (ours only). Top row (a): divergence geometry; circle denotes KL and square denotes $\ell_2$, with $\Delta = \ell_2 - $ KL. Bottom row (b): opponent observation; circle denotes obs $= (s, w, r)$ and square denotes obs $= (s, w)$, with $\Delta = $ obs $= (s, w) - $ obs $= (s, w, r)$. Columns report (left) WRR (higher is better), (middle) DRIFT (lower is better), and (right) GAP (lower is better).

