# OpenReview forum: "Time-Consistent Robust Multi-Objective Reinforcement Learning via a Bellman–Isaacs Weight-Adversary Recursion"
_ICML.cc/2026/Conference — ICML 2026 regular_

### Official Review · Reviewer_XJwn · 2026-03-03

**Soundness:** 3
**Presentation:** 3
**Significance:** 3
**Originality:** 3
**Overall Recommendation:** 4
**Confidence:** 1

**Summary:**

This paper explores how to keep multi-objective AI reliable when its goals keep shifting. Instead of treating the AI's "preference" (what it cares about most) as a fixed setting, the authors treat it as an active enemy that changes its mind at every single step to try and ruin the AI's performance.

To handle this, they built a new math formula where the AI prepares for the absolute worst-case goal change at every turn. They also added a "drift control" rule (using Bregman-divergence) to make sure these goal changes aren't too wild or unrealistic. They proved the math actually settles on a stable solution, created both simple and deep-learning versions of the solver, and even built a stress-test system where they train independent "enemy" AIs to try and find the AI's weaknesses. In tests, this method successfully protected the AI's scores against aggressive goal-shifting while letting users tune exactly how much "drift" they want to allow.

**Compliance With Llm Reviewing Policy:**

Affirmed.

**Final Justification:**

The rebuttal have addressed my concern and I will maintain my positive score.

**Key Questions For Authors:**

See Weakness

**Limitations:**

yes

**Strengths And Weaknesses:**

**Strength**
Instead of just giving the AI a goal, they treat the goal's "priority" as a smart enemy. This enemy changes what the AI should focus on at every single step. To keep it realistic, they added a "switching fee" so the enemy can’t just flip the goals around randomly without a cost.

**Weakness **
The "enemy" in this paper is basically a mind-reader—it gets to see exactly what happened and what the reward was before changing the goal. This might be too harsh compared to the real world, and they didn't test what happens if the enemy has to move first.

They only tested this on games with just two goals and a limited set of choices. They haven't proven this works when you have 5 or 10 different goals or infinite choices, which is where the math usually gets messy.

---

> ### Author Rebuttal · Authors · 2026-03-31
>
> We thank the reviewer for the thoughtful comments. We respond to the three concerns below.
>
> **1. Is the opponent too strong?**
>
> We agree that the main-paper opponent is intentionally strong. The goal is not to assume a literal “mind-reading” adversary. The goal is to model reactive preference shifts: the system first sees the reached state, and the effective priority can then change immediately.
>
> A simple example is safety-critical control. A controller may usually balance progress and efficiency, but after a near-failure state, safety can become the top priority right away. A fixed episode-level weight is too weak for this case, because the priority change happens after the outcome is seen.
>
> This is why our formulation lets the opponent choose the next weight after seeing the realized next state (and, in the strongest setting, the realized reward). Figure 1 and Sec. 3.2–3.3 show why this matters: once the next weight can react to the realized branch, the robustly optimal action can change. So the strong opponent serves two roles. It is the cleanest model of outcome-triggered preference changes, and it is a conservative stress test because a weaker evaluator can miss branch-specific vulnerabilities.
>
> The model is also not “all or nothing.” The switching cost controls how aggressively the weight can move. Proposition 1 shows that when the switching cost becomes very large, the opponent is pushed back toward fixed-weight behavior. So the strong setting is the conservative end of a controlled spectrum, not the only deployment assumption.
>
> **2. Why not test a move-first opponent?**
>
> We agree that this is a reasonable alternative. But it is important to separate two questions.
>
> Our paper studies the case where the next weight is chosen after the outcome is revealed. That is why the minimization sits inside the Bellman backup. If the opponent had to choose its next weight before seeing the outcome, that would be a different and weaker information structure. It would define a different game and a different Bellman operator.
>
> So the key point is not that move-first and our main setting are equivalent. They are not. Move-first is a different robustness notion. Our current theory is for the after-seeing-the-outcome game, and we do not claim that the present theorems already cover the move-first case unchanged.
>
> We did not run a move-first adversary during the rebuttal period. Since it defines a different game/operator from the one analyzed in the paper, we prefer to state that distinction explicitly.
>
> On the empirical side, the current submission does include a weaker-opponent check, though not a move-first one. Specifically, the paper compares a strong opponent that observes $(s_{t+1}, w_t, r_t)$ with a weaker one that observes only $(s_{t+1}, w_t)$. As expected, weakening the opponent’s information makes it less effective. We will clarify this boundary in the revision.
>
> **3. Only two goals / limited choices / what about larger spaces?**
>
> On the theory side, the framework is not limited to a finite set of weight choices. The discussion after Theorem 2 and Appendix B.5 already extend the stationary equilibrium and min–max result to continuous compact $W$.
>
> On the empirical side, we agree that the original experiments were too narrow. During the rebuttal period, we ran 4-objective MO-Lunar-Lander and 6-objective Fruit-Tree.
>
> For these new 4D/6D results, we use the same definitions of WRR and DRIFT as in the paper. The paper only defines GAP for two objectives. Here we report $\mathrm{GAP} := \max_i R_i - \min_i R_i$, where $R$ is the discounted vector return.
>
> 4-objective MO-Lunar-Lander
>
> | Method | WRR ↑ | DRIFT ↓ | GAP ↓ |
> |---|---:|---:|---:|
> | Ours | -3.0473 | 3.1751 | 31.9227 |
> | CN-DQN | -6.7256 | 108.4052 | 40.5880 |
> | Envelope | -5.1989 | 34.6925 | 33.1273 |
> | PCN | -53.5297 | 13.1815 | 88.7153 |
> | Fixed-w | -16.6088 | 126.8366 | 85.9096 |
>
> 6-objective Fruit-Tree
>
> | Method | WRR ↑ | DRIFT ↓ | GAP ↓ |
> |---|---:|---:|---:|
> | Ours | 6.6308 | 3.4498 | 2.5217 |
> | CN-DQN | 4.6773 | 17.2689 | 5.8035 |
> | Envelope | 6.5321 | 3.7428 | 7.7547 |
> | PCN | 3.6501 | 10.2136 | 4.7831 |
> | Fixed-w | 3.9963 | 10.8526 | 2.7528 |
>
> In both 4D and 6D, Ours is best on all three metrics. The 6-objective Fruit-Tree task is also the highest-dimensional benchmark in the MO-Gymnasium suite we use, so it directly addresses the concern that the method might only work in very small objective spaces.
>
> Overall, our intended claim is therefore narrower and clearer: the paper targets robustness to reactive, step-wise preference shifts after outcomes are seen. The strong opponent is a conservative evaluator for that case. Move-first is a useful weaker alternative, but it is a different game. The new 4D and 6D results further show that the method is not tied to only two objectives.

---

> > ### Author Rebuttal · Reviewer_XJwn · 2026-04-02
> >
> > Thanks the authors for the rebuttal. I will keep my positive score.

---

> > > ### Author Response · Authors · 2026-04-06
> > >
> > > Thank you very much for the helpful follow-up and for the careful reading. We are glad that our rebuttal helped clarify the main points. In the final version, we will revise the presentation to address the issues you raised more clearly, especially by improving readability and reducing the amount of effort needed for a first pass through the paper. In particular, we will make the motivation, scope, and empirical takeaways more direct, and we will further simplify the exposition so that the core message is easier to follow.

---

### Official Review · Reviewer_6AMW · 2026-03-11

**Soundness:** 3
**Presentation:** 3
**Significance:** 3
**Originality:** 3
**Overall Recommendation:** 4
**Confidence:** 1

**Summary:**

This paper studies a novel setting of MORL when an opponent has the ability to choose the next time step's weight with a switching cost. The paper theoretically proves that the induced Bellman-Isaacs operator is $\gamma$ contraction and establishes a stationary equilibrium, and derives an auditable certificates that convert Bellman residuals into robust-performance guarantees. This paper also introduces a novel evaluation protocol called BR-K.

**Compliance With Llm Reviewing Policy:**

Affirmed.

**Final Justification:**

The rebuttal has addressed my concerns.

**Key Questions For Authors:**

1. **Practical implications:**
The paper presents an interesting theoretical formulation of MORL as an adversarial optimization problem. Could the authors elaborate on the practical implications of this formulation? In particular, how might this framework be applied in real-world MORL scenarios, and what types of applications would benefit most from modeling MORL with an adversarial opponent?

2. **Empirical evaluation:**
The current empirical evaluation appears relatively limited. Could the authors provide additional experiments on more environments or benchmarks to further demonstrate the effectiveness and generality of the proposed framework?

**Limitations:**

Yes, this paper includes a limitation discussion.

**Strengths And Weaknesses:**

**Strengths**

**Soundness:**
The paper proposes a novel framework that reformulates multi-objective reinforcement learning (MORL) as an opponent-based robust optimization problem. The work provides detailed theoretical analysis and establishes formal results under this formulation. The comprehensive theoretical treatment strengthens the technical soundness of the paper and provides useful insights into the robustness properties of the proposed framework.

**Presentation:**
The paper is well structured and clearly presents the motivation behind the proposed formulation. The authors explain how the MORL problem can be reframed through an adversarial perspective, and the progression from problem formulation to theoretical analysis is generally easy to follow.

**Significance:**
The paper studies robustness in MORL under adversarial settings, which is an important problem as reinforcement learning systems are increasingly deployed in uncertain and potentially adversarial environments.

**Originality:**
To the best of my knowledge, modeling MORL as an adversarial optimization problem involving an opponent is a novel perspective. The theoretical analysis of this formulation provides new insights into robustness in multi-objective reinforcement learning.

---

**Weaknesses**

**Empirical validation:**
The paper focuses heavily on theoretical analysis, but the empirical evaluation is relatively limited. Additional experiments or more diverse benchmarks would help demonstrate the practical effectiveness of the proposed framework.

**Practical applicability:**
While the adversarial formulation is theoretically interesting, the practical implications for real-world MORL applications are not fully clear. Additional discussion on how this framework could be applied in practical scenarios would improve the paper.

---

> ### Author Rebuttal · Authors · 2026-03-31
>
> We thank the reviewer for the helpful feedback. We respond to the two concerns below.
>
> **1. Practical implications**
>
> We agree that this part should be more concrete. Our goal is not to claim that every real system has a literal malicious user who changes preferences at every step. The adversary should instead be viewed as a conservative model for reactive preference shifts: the state changes first, and the effective priority changes immediately after that.
>
> A concrete example is autonomous landing. Early in an episode, the controller may balance landing progress, smooth motion, and fuel use. If the vehicle suddenly tilts too much or drops too fast, safety must become the top priority right away. If the state stabilizes again, fuel efficiency can matter more. A single episode-level weight cannot represent this kind of process, because the priority change happens after the new state is observed.
>
> The same pattern appears in other robotics settings. A robot may normally balance task progress, energy, and collision risk, but after a near-collision or low-battery event it may need to shift immediately toward safety or recovery. A similar idea also applies to recommendation systems: the platform may balance short-term engagement, diversity, and long-term satisfaction, but after repeated skips or clear fatigue signals, the desired trade-off can shift away from clicks and toward safer long-term behavior.
>
> So the practical message is not “there is always a malicious opponent.” It is that our formulation gives a conservative way to evaluate policies when trade-offs can change along the trajectory in response to outcomes.
>
> **2. Empirical evaluation**
>
> We agree that the original empirical section was too narrow. During the rebuttal period, we ran two additional higher-dimensional benchmarks: 4-objective MO-Lunar-Lander and 6-objective Fruit-Tree.
>
> For these new 4D/6D results, we use the same definitions of WRR and DRIFT as in the paper. The paper only defines GAP for two objectives. Here we report $\\mathrm{GAP} := \\max_i R_i - \\min_i R_i$, where $R$ is the discounted vector return.
>
> 4-objective MO-Lunar-Lander
>
> | Method | WRR ↑ | DRIFT ↓ | GAP ↓ |
> |---|---:|---:|---:|
> | Ours | -3.0473 | 3.1751 | 31.9227 |
> | CN-DQN | -6.7256 | 108.4052 | 40.5880 |
> | Envelope | -5.1989 | 34.6925 | 33.1273 |
> | PCN | -53.5297 | 13.1815 | 88.7153 |
> | Fixed-w | -16.6088 | 126.8366 | 85.9096 |
>
> Our method is best on all three metrics. It gets the highest WRR while also having the lowest DRIFT and the lowest GAP.
>
> 6-objective Fruit-Tree
>
> | Method | WRR ↑ | DRIFT ↓ | GAP ↓ |
> |---|---:|---:|---:|
> | Ours | 6.6308 | 3.4498 | 2.5217 |
> | CN-DQN | 4.6773 | 17.2689 | 5.8035 |
> | Envelope | 6.5321 | 3.7428 | 7.7547 |
> | PCN | 3.6501 | 10.2136 | 4.7831 |
> | Fixed-w | 3.9963 | 10.8526 | 2.7528 |
>
> The same pattern holds in 6D. Ours again has the best WRR, the lowest DRIFT, and the lowest GAP. Among the MO-Gymnasium tasks we currently use, this Fruit-Tree benchmark is also the highest-dimensional one, so it is a direct stress test beyond the original 2D setting.
>
> These results are not in the current submission, and we will include them together with a clearer practical discussion in the revision.

---

> > ### Author Rebuttal · Reviewer_6AMW · 2026-04-01
> >
> > Thanks for the rebuttal. I am maintaining my positive rating for this paper.

---

> > > ### Author Response · Authors · 2026-04-06
> > >
> > > Thank you very much for the helpful follow-up and for the careful reading. We are glad that our rebuttal helped clarify the main points. In the final version, we will revise the presentation to address the issues you raised more clearly, especially by improving readability and reducing the amount of effort needed for a first pass through the paper. In particular, we will make the motivation, scope, and empirical takeaways more direct, and we will further simplify the exposition so that the core message is easier to follow.

---

### Official Review · Reviewer_thyg · 2026-03-13

**Soundness:** 3
**Presentation:** 3
**Significance:** 3
**Originality:** 4
**Overall Recommendation:** 4
**Confidence:** 2

**Summary:**

This paper studies robust MORL under time-consistent, step-wise preference shifts by modeling the weight vector as an adversarial control chosen after each transition, which leads to a Bellman–Isaacs recursion. It also proposes practical tabular/deep solvers and a BR-K evaluation protocol using multiple independently trained best-response adversaries.

**Compliance With Llm Reviewing Policy:**

Affirmed.

**Final Justification:**

The problem formulation is clear and novel. It proposes a sound solution built on a solid theoretical foundation. The authors’ rebuttal has resolved my concerns. I still think it would be better if the benchmarks were more realistic, with higher task complexity and more practically motivated preference transitions, but I would not see this as a major problem given the additional experiments on the 4-objective MO-Lunar-Lander and 6-objective Fruit-Tree included in the rebuttal. Overall, I will keep my positive score.

**Key Questions For Authors:**

1. Can the authors provide more concrete real-world scenarios where this adversarial formulation is practically necessary?
2. Since BR-K still relies on learned best-response adversaries, how sensitive are the conclusions to the adversary training budget and the number of restarts $K$? Would the rankings remain stable with stronger BR training?
3. The experiments appear focused on a small set of two-objective benchmarks. How well does the method scale to richer environments or higher-dimensional MORL settings, where the inner minimization and representation of $Q(s,w,a)$ may become more challenging?

**Limitations:**

Yes.

**Strengths And Weaknesses:**

**Strengths**
1. The problem formulation is clear and novel: unlike episode-level robust MORL with a fixed adversarial weight, the proposed step-wise formulation captures state-reactive preference shifts and enforces time consistency.
2. The theoretical side is solid, with contraction and unique fixed-point results for the exact operator.
3. The BR-K protocol is also a sensible addition, since a single learned adversary may be too weak.

**Weaknesses**
1. While step-wise adversarial weight model is a reasonable worst-case formulation, the paper would benefit from more concrete real-world scenarios where such a powerful adversary is practically necessary.
2. The empirical scope is limited. Experiments are conducted on a small set of two-objective benchmarks, so it remains unclear how well the approach scales to richer or higher-dimensional MORL settings.

---

> ### Author Rebuttal · Authors · 2026-03-31
>
> We thank the reviewer for the helpful comments. We address the three concerns below.
>
> **1. Concrete real-world scenarios**
>
> We agree this should be made more concrete. The key point is not that every deployment has a literal adversary. The key point is that, in many systems, the state changes first and the effective priority changes right after that.
>
> A concrete example is autonomous landing. Early in an episode, the controller may balance landing progress, fuel use, and smooth motion. If the vehicle suddenly tilts too much or descends too fast, safety must become the top priority immediately. If the state stabilizes again, fuel efficiency can matter more. A single episode-level weight cannot capture this kind of state-reactive shift.
>
> The same pattern appears in mobile robotics. A robot may normally balance task progress, energy, and collision risk. After a near-collision or a low-battery event, the effective priority can shift immediately toward safety or recovery. A similar idea also fits recommendation systems, where repeated skips or fatigue signals can shift the trade-off away from short-term engagement and toward long-term satisfaction.
>
> **2. Sensitivity of BR-K to budget and K**
>
> We agree this is important. To answer it directly, we stress-tested BR training on a fixed Timespeed/Ours checkpoint and varied both the number of restarts and the BR training budget. Throughout this check, we follow the paper’s BR-K rule: for each setting, we select the adversary with the lowest mean WRR against the frozen agent.
>
> Worst-of-K mean WRR on the same checkpoint
>
> | K | 3 | 5 | 10 |
> |---|---:|---:|---:|
> | WRR ↑ | -62.7043 | -62.7393 | -62.7412 |
>
> This changes only slightly from K=3 to K=10, so the worst-case estimate saturates quickly.
>
> BR budget on the same checkpoint, with K=5 fixed
>
> | BR budget | WRR ↑ | DRIFT ↓ | GAP ↓ |
> |---|---:|---:|---:|
> | 200k | -62.7393 | 0.5596 | 81.3547 |
> | 500k | -62.0274 | 0.4649 | 79.7071 |
>
> Increasing the BR budget from 200k to 500k changes the estimate only modestly and does not qualitatively alter the conclusion. This is a fixed-checkpoint stress test rather than a full cross-method re-ranking under stronger BR training, but it indicates that the worst-case estimate itself is already fairly stable at the scale used in the paper.
>
> **3. Richer environments and higher-dimensional objectives**
>
> We also agree that stronger evidence beyond the original 2-objective benchmarks is needed. During the rebuttal period, we ran 4-objective MO-Lunar-Lander and 6-objective Fruit-Tree.
>
> For these new 4D/6D results, we use the same definitions of WRR and DRIFT as in the paper. The paper only defines GAP for two objectives. Here we report $\\mathrm{GAP} := \\max_i R_i - \\min_i R_i$, where $R$ is the discounted vector return.
>
> 4-objective MO-Lunar-Lander
>
> | Method | WRR ↑ | DRIFT ↓ | GAP ↓ |
> |---|---:|---:|---:|
> | Ours | -3.0473 | 3.1751 | 31.9227 |
> | CN-DQN | -6.7256 | 108.4052 | 40.5880 |
> | Envelope | -5.1989 | 34.6925 | 33.1273 |
> | PCN | -53.5297 | 13.1815 | 88.7153 |
> | Fixed-w | -16.6088 | 126.8366 | 85.9096 |
>
> 6-objective Fruit-Tree
>
> | Method | WRR ↑ | DRIFT ↓ | GAP ↓ |
> |---|---:|---:|---:|
> | Ours | 6.6308 | 3.4498 | 2.5217 |
> | CN-DQN | 4.6773 | 17.2689 | 5.8035 |
> | Envelope | 6.5321 | 3.7428 | 7.7547 |
> | PCN | 3.6501 | 10.2136 | 4.7831 |
> | Fixed-w | 3.9963 | 10.8526 | 2.7528 |
>
> In both 4D and 6D, Ours is best on all three metrics. Fruit-Tree is also the highest-dimensional benchmark in the MO-Gymnasium suite we use, so the 6D result is a genuine high-dimensional stress test rather than a small extension of the original setup.
>
> These additional results are not in the current submission, and we will include both the BR-sensitivity analysis and the new higher-dimensional results in the revision.

---

> > ### Author Rebuttal · Reviewer_thyg · 2026-04-02
> >
> > Thanks for the rebuttal. My concerns have been resolved, and I will keep my positive score.
> >
> > Regarding Question 1 and 3, I believe the paper's impact would be even greater if it demonstrated strong performance on more realistic benchmarks with higher task complexity and more practically motivated preference transitions.

---

> > > ### Author Response · Authors · 2026-04-06
> > >
> > > Thank you very much for the helpful follow-up and for the careful reading. We are glad that our rebuttal helped clarify the main points. In the final version, we will revise the presentation to address the issues you raised more clearly, especially by improving readability and reducing the amount of effort needed for a first pass through the paper. In particular, we will make the motivation, scope, and empirical takeaways more direct, and we will further simplify the exposition so that the core message is easier to follow.

---

### Official Review · Reviewer_nncB · 2026-03-20

**Soundness:** 3
**Presentation:** 3
**Significance:** 2
**Originality:** 3
**Overall Recommendation:** 5
**Confidence:** 4

**Summary:**

This paper studies a time-consistent robust multi-objective reinforcement learning (MORL) setting, where an adversary selects a weight vector at each transition step to scalarize the reward vector. Compared to prior work, this represents a stronger adversarial setup, as the adversary can adapt weights dynamically over time, while incurring a switching cost to penalize rapid changes.

The problem is formulated as a discounted zero-sum stochastic game over the joint space of (state, weight). To solve this game, the authors introduce a Bellman–Isaacs operator and establish its key theoretical properties, including contraction, enabling value-iteration-style solutions.

Building on this foundation, the paper proposes two practical algorithms:
- A tabular method using a discretized weight grid.
- A deep learning approach leveraging mirror-prox methods.

Empirically, the proposed methods are evaluated against K-best-response adversaries trained against a fixed agent policy. Experiments on 2D MO-Gym environments show that the approach outperforms preference-conditioned MORL baselines.

**Compliance With Llm Reviewing Policy:**

Affirmed.

**Final Justification:**

The authors have addressed my key questions with additional experiments on continuous adversary and high-dimensional objectives (please see strengths and weaknesses). Therefore, I am increasing my score by 1.

**Key Questions For Authors:**

[1] Continuous adversary optimization

In Algorithm 3 (BR-K evaluation), the adversary is optimized using a DQN-style approach over a discrete weight space.

Would it be possible to use continuous-control methods (e.g., TD3 or DDPG) for adversary optimization to evaluate robustness over a continuous weight space?

[2] Scalability to higher-dimensional objectives

The experiments focus on two-objective settings.

Could the authors evaluate the proposed method in higher-dimensional objective spaces to better understand its scalability?

**Limitations:**

yes

**Strengths And Weaknesses:**

**Strengths**

The paper is generally well-written and clearly structured.

The work addresses a meaningful extension of MORL by introducing time-consistent adversarial robustness, which is more challenging than static formulations.

The formulation as a stochastic game is well-motivated. The analysis of the Bellman–Isaacs operator (e.g., contraction properties) is rigorous and provides solid grounding for the algorithms.

**Weaknesses**

Limited empirical scope: Experiments are restricted to two-objective settings, leaving scalability to higher-dimensional objectives unclear.

Evaluation limitations: Robustness is primarily assessed using a discretized weight grid, which may not fully capture performance in continuous weight spaces.

---

> ### Author Rebuttal · Authors · 2026-03-31
>
> We thank the reviewer for the constructive questions. We respond to the two points below.
>
> **1. Continuous adversary optimization**
>
> Yes. The current BR-K evaluator in the deep experiments uses a discrete weight grid, so this is a fair question. At the same time, the formulation itself is not restricted to discrete weights. In the paper, the discussion after Theorem 2 and Appendix B.5 already extend the equilibrium picture to continuous compact $W$. So the discrete BR evaluator is an implementation choice, not a limitation of the underlying objective.
>
> To answer the question directly, during the rebuttal period we ran a continuous best-response adversary using TD3 on DeepSeaTreasure. The adversary now outputs $w_{t+1}$ continuously on the simplex, instead of choosing from a finite grid. We kept the same post-transition observation and the same stage objective as in the paper; only the adversary action space changed.
>
> Since DeepSeaTreasure is still a 2-objective setting, we report the same three metrics used in the paper:
> - WRR: adversarial weighted return. Higher is better.
> - DRIFT: cumulative preference switching. Lower is better.
> - GAP: spread between the two discounted objective returns. Lower is better.
>
> Continuous TD3 adversary on DeepSeaTreasure
>
> | Method | WRR ↑ | DRIFT ↓ | GAP ↓ |
> |---|---:|---:|---:|
> | Ours | -1.0000 | 0.5821 | 1.7000 |
> | Fixed-w | -5.8520 | 0.6931 | 5.8520 |
>
> Under a continuous TD3 adversary, Ours remains clearly stronger than Fixed-w: WRR improves by 4.8520, DRIFT decreases by 0.1110, and GAP decreases by 4.1520. This directly answers the main concern: moving from a discrete evaluator to a continuous evaluator does not overturn the conclusion.
>
> **2. Scalability to higher-dimensional objectives**
>
> We agree that the original experiments were too concentrated on two-objective settings. During the rebuttal period, we also ran 4-objective MO-Lunar-Lander and 6-objective Fruit-Tree.
>
> For these new 4D/6D results, we use the same definitions of WRR and DRIFT as in the paper. The paper only defines GAP for two objectives. Here we report $\\mathrm{GAP} := \\max_i R_i - \\min_i R_i$, where $R$ is the discounted vector return.
>
> 4-objective MO-Lunar-Lander
>
> | Method | WRR ↑ | DRIFT ↓ | GAP ↓ |
> |---|---:|---:|---:|
> | Ours | -3.0473 | 3.1751 | 31.9227 |
> | CN-DQN | -6.7256 | 108.4052 | 40.5880 |
> | Envelope | -5.1989 | 34.6925 | 33.1273 |
> | PCN | -53.5297 | 13.1815 | 88.7153 |
> | Fixed-w | -16.6088 | 126.8366 | 85.9096 |
>
> In 4D, Ours is best on all three metrics.
>
> 6-objective Fruit-Tree
>
> | Method | WRR ↑ | DRIFT ↓ | GAP ↓ |
> |---|---:|---:|---:|
> | Ours | 6.6308 | 3.4498 | 2.5217 |
> | CN-DQN | 4.6773 | 17.2689 | 5.8035 |
> | Envelope | 6.5321 | 3.7428 | 7.7547 |
> | PCN | 3.6501 | 10.2136 | 4.7831 |
> | Fixed-w | 3.9963 | 10.8526 | 2.7528 |
>
> The same pattern holds in 6D. Ours again has the best WRR, the lowest DRIFT, and the lowest GAP. Fruit-Tree is the highest-dimensional benchmark in the MO-Gymnasium suite we use, so this is a direct scalability check rather than another small variation of the original 2D tasks.
>
> These additional results are not in the current submission, and we will add them in the revision. We will also clarify there that continuous-control best responses are fully compatible with our formulation, while the original discrete BR-K protocol was chosen mainly for evaluator stability and comparability.

---

> > ### Author Rebuttal · Reviewer_nncB · 2026-04-02
> >
> > The authors have addressed my key questions. I will increase my score.

---

> > > ### Author Response · Authors · 2026-04-06
> > >
> > > Thank you very much for the helpful follow-up and for letting us know that our rebuttal addressed your key concerns. We are glad that the additional clarification was useful. We will make sure these points are incorporated more clearly in the revised version.
> > >
> > > We would also appreciate it if, when convenient, you could update the official score in the system so that it reflects your current assessment.

---

### Decision · Program_Chairs · 2026-04-30

**Decision:**

Accept (regular)

**Comment:**

The paper proposes a novel formulation of robust multi-objective reinforcement learning in which preference weights are selected step-wise by an adversary rather than fixed once per episode. Reviewers generally agreed that this is a clear and meaningful conceptual advance over prior episodic robust MORL formulations.

The main concerns were about the scope of experiments. Multiple reviewers noted that the original experiments were on only two-objective benchmarks, leaving open the scalability. The authors’ rebuttal addressed the concerns. In particular, they added experiments on more benchmarks including 4-objective MO-Lunar-Lander and 6-objective Fruit-Tree.

Overall, I recommend acceptance for this paper.